# SORTeD Rashomon Sets of Sparse Decision Trees: Anytime Enumeration

**Elif Arslan**
Delft University of Technology, Netherlands
E.Arslan-1@tudelft.nl

**Jacobus G. M. van der Linden**
Delft University of Technology, Netherlands
J.G.M.vanderLinden@tudelft.nl

**Serge Hoogendoorn**
Delft University of Technology, Netherlands
S.P.Hoogendoorn@tudelft.nl

**Marco Rinaldi**
Delft University of Technology, Netherlands
M.Rinaldi@tudelft.nl

**Emir Demirović**
Delft University of Technology, Netherlands
E.Demirovic@tudelft.nl

## Abstract

Sparse decision tree learning provides accurate and interpretable predictive models that are ideal for high-stakes applications by finding the single most accurate tree within a (soft) size limit. Rather than relying on a single "best" tree, Rashomon sets—trees with similar performance but varying structures—can be used to enhance variable importance analysis, enrich explanations, and enable users to choose simpler trees or those that satisfy stakeholder preferences (e.g., fairness) without hard-coding such criteria into the objective function. However, because finding the optimal tree is NP-hard, enumerating the Rashomon set is inherently challenging. Therefore, we introduce SORTD, a novel framework that improves scalability and enumerates trees in the Rashomon set in order of the objective value, thus offering anytime behavior. Our experiments show that SORTD reduces runtime by up to two orders of magnitude compared with the state of the art. Moreover, SORTD can compute Rashomon sets for any separable and totally ordered objective and supports post-evaluating the set using other separable (and partially ordered) objectives. Together, these advances make exploring Rashomon sets more practical in real-world applications.

## 1 Introduction

Decision trees are widely regarded as one of the most interpretable models, as their decision paths are easy to follow and understand. While it is commonly believed that interpretability comes at the cost of accuracy, recent findings suggest that in some applications this trade-off may be negligible [1]. This makes decision trees even more appealing in practice, specifically for high-stakes domains.

Decision trees have been extensively studied, with early popular approaches such as CART [2] and C4.5 [3] focusing on greedy top-down induction. Because the size of a tree correlates with its comprehensibility [4], recent research has also examined *optimal* or *sparse* decision trees that maximise performance for a given size limit, thereby obtaining a better accuracy–interpretability trade-off [5, 6]. Consequently, the literature presents a variety of approaches to compute such optimal trees, including mixed-integer programming [7–9], Boolean satisfiability [10–12], constraint programming [13], branch-and-bound [14], and dynamic programming [15–18].

39th Conference on Neural Information Processing Systems (NeurIPS 2025).

These methods share the property of returning a single decision tree that is constructed to perform well with respect to a loss function. However, there are typically many *approximately equally good—but possibly very different—decision-tree models* for a given learning problem. This phenomenon is known as the *Rashomon effect* [19], and the set of all models within a tolerance of the globally optimal loss value is called the *Rashomon set*.

Rashomon sets offer several advantages over relying on a single best model. In explainable AI, they enable the discovery of simpler models [20, 21], support diverse counterfactual explanations [22], address underspecification [23], also in explanations [24], and improve variable-importance analysis [25]. In addition, they allow evaluation of criteria that are difficult to encode directly into the objective function—such as fairness or robustness—by analysing properties across the set. In recidivism prediction, Fisher et al. [26] examined the influence of bias-associated variables, and Marx et al. [27] explored how similar models might yield conflicting predictions. Rashomon sets also provide a more flexible interface for stakeholders, enabling them to select models that best align with their domain-specific constraints. In healthcare, a diverse set of models obtained from the Rashomon set was used to support informed decision-making [28, 29].

Benefiting from the Rashomon set requires efficient evaluation, as the set size can be prohibitively large (with only ten features and a depth budget of four, it may exceed $10^{12}$ trees). Practical use cases often require finding the most accurate trees, while also satisfying structural constraints (e.g., limits on tree size or mandatory/forbidden features) or attaining favourable scores on complementary objectives (e.g., fairness). This search task would be greatly simplified if candidate trees were generated in non-decreasing order of their objective value, preventing users from spending substantial time to examine the entire set to locate the desired models.

The current state-of-the-art for obtaining the Rashomon set of sparse decision trees [30] returns every tree whose performance falls within a user-chosen tolerance of the best model (e.g., 5% above the optimum) but the output is not ordered by the objective. If the search is terminated early, e.g., because of a time-out, there is no guarantee that the returned set contains the actually best models according to the objective. Furthermore, when the appropriate margin is uncertain, it is often more practical to select a larger value, which can result in the construction of an extremely large set; conversely, if a fixed number of top-ranked models is of interest, there is no systematic mechanism to terminate enumeration once that quantity has been reached.

To address these concerns, we propose a novel framework, **SORTD** (**So**rted **R**ashomon Sets of **T**rees using **D**ynamic Programming). In doing so, we achieve three contributions. First, we compute the Rashomon set *in order*: trees with the best objective values are generated first. Ordered generation enables early termination when a specified number of high-quality models has been obtained, hence providing an anytime Rashomon set property. Our experiments show that having access to the best models early can speed up downstream evaluation tasks such as variable-importance analysis.

Our second contribution is *improved scalability in the Rashomon set calculation*. To reduce runtime, SORTD incorporates a specialised algorithm for trees of depth two or less. This design allows SORTD to scale well with both the number of features and the depth budget. In our evaluation on a variety of benchmark classification datasets, we show that SORTD significantly outperforms the state-of-the-art, achieving speed-ups of up to *two orders of magnitude*.

Finally, our third contribution is *providing a general framework that computes Rashomon sets for separable and totally ordered objectives and evaluates them with any separable and partially ordered one*. We demonstrate this by enumerating the Rashomon set also for regression trees in addition to classification, and evaluating the Rashomon set of decision trees using an additional fairness objective. Together, these contributions make SORTD a practical and scalable tool for decision tree training, evaluation, and selection.

The rest of the paper is organized as follows: Sec. 2 reviews related work; Sec. 3 covers preliminaries; Sec. 4 details the Rashomon set calculation; and Sec. 5 presents the experiments.

## 2 Related work

**Methods** Rashomon set computation has recently been explored for risk score models [29, 31], additive models [20, 24, 32], rule sets [33, 34], random forest [24], and kernel ridge regression [24]. For sparse decision trees—the focus of this work—the only dedicated approach is TreeFARMS [30],

which enumerates every tree within a given user-chosen tolerance (the "Rashomon multiplier") but does not preserve a global ordering of trees with respect to the objective. However, this user tolerance is rarely known a priori: an overly large value may result in generating billions of trees, which is memory and time intensive; while a small value may return too few trees. Additionally, TreeFARMS is tailored to classification, and extension to other optimization tasks is non-trivial.

On the contrary, our method produces solutions iteratively in non-decreasing order, allowing the algorithm to stop as soon as a target number of high-quality trees is reached without relying on an accurately tuned tolerance. This gives SORTD an anytime behavior: stopping the search at any time yields a Rashomon set. Furthermore, a specialised depth-two solver reduces runtime and improves scalability with both the number of features and the depth budget. Finally, SORTD handles any separable and totally ordered loss function and supports post-hoc evaluation of separable and partially ordered objectives (e.g., multi-objective optimization), hence increasing flexibility in learning and evaluation.

**Decision trees**    Early decision tree induction methods, such as AID [35] for recursive regression analysis, and CHAID [36] for classification, use top-down induction to infer the next best split. The two most popular approaches, CART [2] and C4.5 [3], share this paradigm. While these methods typically yield good results, their greedy nature may yield models that are arbitrarily larger than optimal [37]. Indeed, provably optimal trees that are obtained through exhaustive search on average obtain a better size-accuracy, and hence interpretability-accuracy, trade-off than greedy approaches [6, 7]. Although finding such optimal trees is NP-hard [38], the problem remains tractable for a limited number of features and small tree-size limits [39], and recent dynamic programming (DP) approaches can typically find optimal trees of limited size for real-world datasets in seconds [e.g., 16, 40].

Unlike these approaches, our work aims not to find a single best tree, but the set of all good trees. A key advantage is that this set can be explored to find optimal solutions for other objectives or constraints that are harder to optimize directly. For example, while Demirović et al. [41] develop a specialized DP algorithm for non-linear metrics such as F1-score, Xin et al. [30] obtain the optimal F1-score tree from the Rashomon set based on optimizing accuracy. Similarly, rather than building a custom method for each objective or constraint, such as for example, a demographic parity fairness constraint [42], we explore the Rashomon set instead to find trees that are both accurate and fair.

## 3    Preliminaries

**Notation**    Given a set of binary *features* $F$ and a set of *labels* $K$, a sample $(x_i, k_i)$ is a pair of feature vector $x_i \in \{0, 1\}^{|F|}$ and label $k_i \in K$. A dataset $D = \{(x_i, k_i)\}_i$ is the set of samples that can be used to train a prediction model. Since we assume that the features are binary, we use $D(f)$ to indicate the set of samples where $f$ is satisfied and $D(\bar{f})$ is the set of samples where $f$ is not satisfied.

**Sparse tree objective**    A binary tree is a function $T : \{0, 1\}^{|F|} \rightarrow K$ that recursively maps a feature vector $x$ to a predicted label $\hat{k}$. Starting with the root node, each internal node in $T$ sends an instance left or right when its specified binary feature test is satisfied in $x$ or not. The final leaf node then provides its assigned label as the return value. The optimization task in this work considers finding trees that optimize a given objective function. In Appendix A.9, we consider arbitrary *separable and totally ordered* objectives [40], but in the main text, we limit our discussion for brevity to finding optimal sparse classification trees, for which we need to find the tree that minimizes:

$$C(T, D) = \frac{1}{|D|} \sum_{(x,k) \in D} \mathbb{1}[T(x) \neq k] + \lambda N(T). \tag{1}$$

This equation penalizes each misclassification and additionally, the number of leaf nodes $N(T)$ by a complexity cost $\lambda$ [43]. Given Eq. (1), $T^* = \text{argmin}_{T \in \mathcal{T}(d)} C(T, D)$ finds the optimal tree from the set of all trees $\mathcal{T}(d)$ of maximum depth $d$.

**Rashomon set**    Given a Rashomon multiplier $\varepsilon$, the Rashomon set is the set of trees with an objective value within the Rashomon bound $\theta(T^*, D, \varepsilon) = (1 + \varepsilon)C(T^*, D)$. We obtain the Rashomon set:

$$R(T^*, D, \varepsilon) = \{T \in \mathcal{T}(d) \mid C(T, D) \leq \theta(T^*, D, \varepsilon)\}. \tag{2}$$

**Depth-two subroutine** A specialized subroutine for finding optimal trees of depth two was proposed by Demirović et al. [16]. It has a significant computation advantage by exploiting precomputed frequency counts instead of repeatedly recursively splitting the dataset. Taking binary classification as example, with $D^+$ the set of positive samples, then $Q^+(f_i) = |\{(x,k) : (x,k) \in D^+(f_i)\}|$ and $Q^+(f_i, f_j) = |\{(x,k) : (x,k) \in D^+(f_i) \cap D^+(f_j)\}|$ are the number of occurrences of feature $f_i$, and of $f_i$ and $f_j$ combined respectively in the positive samples, which can be counted efficiently by looping over sparse representations of the feature vectors. The frequency counts for other possible feature inclusions or exclusions of features $f_i$ and $f_j$ in a depth-two tree are calculated implicitly using only these two definitions. E.g., $Q^+(f_i, \bar{f}_j) = Q^+(f_i) - Q^+(f_i, f_j)$ represents the number of instances that satisfy feature $f_i$, but not $f_j$. The frequency counts $Q^-$ for negative labels can be calculated analogously. Combining both positive and negative frequency counts allows us to directly compute misclassification scores for all possible depth-two trees.

## 4 In-order Rashomon set calculation

### 4.1 High-level idea

Given a depth budget, we construct the Rashomon set by exploring decision trees in ascending (i.e., best-first) order of their objective values. We first identify the optimal tree and set the Rashomon bound (tolerance of the best model). We then iteratively build the Rashomon set by maintaining sorted lists of solutions for each node in the search tree. Note that our contribution lies in this second phase: efficiently identifying and ordering these additional solutions.

To compute the Rashomon set, we use a search tree. Each search node maintains a sorted solution list, starting with the optimal one(s), followed by the suboptimal solutions in order of their objective value. A search node contains a helper node for each possible split on all features $f \in F$, and one for creating a leaf node. Each helper node —branching or leaf— of the search node also maintains its own sorted solution list and a pointer to its next unprocessed solution with minimum objective value. Leaf nodes contribute a single solution while branching nodes generate multiple.

*In essence, during Rashomon set computation, a search node either repeatedly returns its best next solution from its sorted list or explores new candidates through its helper nodes when no further solution is available. In this exploration phase, the node selects the best next solution among its helpers, and the chosen helper prepares its next candidate if one exists. The search node then returns this selected solution. This process continues until the Rashomon bound or the Rashomon set size is reached. This lazy enumeration strategy computes new solutions only when needed, thereby reducing computation time.*

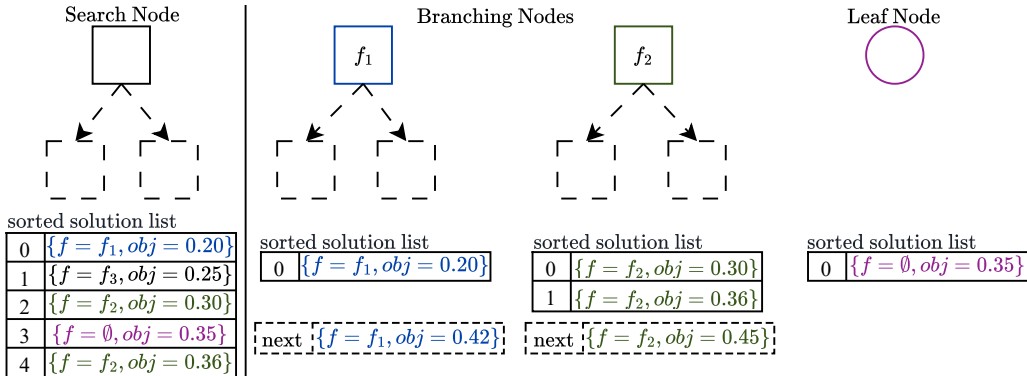

Figure 1: Search tree structure. The left-most node is the current search node with its sorted solution list. The middle nodes are branching nodes with features $f_1$ and $f_2$. The right-most is a leaf node.

**Example 1.** *Fig. 1 shows a search node with three helper nodes: two of its branching nodes and one leaf node. Its solution list aggregates the solutions from the helpers. The leaf node's only solution is already included. The branching node with feature $f_1$ has the next minimum solution value $0.42$.*

Fig. 2 shows how a branching node computes its next solution by combining the best solutions from its left and right child nodes. It computes the Cartesian sum $\{a + b \mid a \in A, b \in B\}$ (see, e.g., [44,

45]), but iteratively and in sorted order, while the sets $A$ and $B$ (the solution lists of the child nodes) at the same time are also iteratively generated (and also in-order). The best solution of this branch node is formed by pairing the child search nodes' best solutions (their solutions with index 0). To find the next-best solution, we increment either the left or right index. The lower-valued combination is selected as the next solution, and the other is stored in the candidates queue. In each step, the next solution is either the minimum valued candidate or a new combination obtained by incrementing one of the indices of the current next solution.

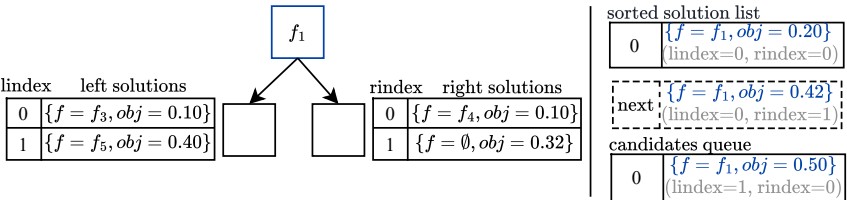

Figure 2: Next solution calculation in a branching node.

**Example 2.** *In Fig. 2, after adding the best solution of value* $0.10 + 0.10 = 0.20$ *(lindex = 0, rindex = 0), two new options with values* $0.10 + 0.32 = 0.42$ *(lindex = 0, rindex = 1) and* $0.40 + 0.10 = 0.50$ *(lindex = 1, rindex = 0) are considered. The former becomes the next solution, and the latter is stored in the candidates queue.*

## 4.2 Algorithm

Phase 1 of our algorithm calculates the optimal tree given a dataset, and a depth budget and meanwhile also caches optimal solutions to subproblems. In Phase 2—the focus of our contribution—we create a search tree over subproblems, with each search node holding a sorted list of computed solutions (so far), and the value of the next best solution (initially set to the cached optimal partial solutions). Phase 2 then iteratively requests the next best solution from the root search node, which recursively constructs it by requesting the next best solutions from its child search nodes (see Appendix A.1).

Alg. 1 shows how a search node computes its next best solution. First, the helper node with the smallest next solution value $node.next$ is found (Line 1). A solution group $sol$ is then created to collect all solutions with the same objective value (Line 2). If the helper node is a branching node, additional solutions with the same value are retrieved from its candidates queue $CQ$ (Lines 4-5). This queue is a heap of tuples $(sol.value, lindex, rindex)$, sorted by $sol.value$, while $lindex$ and $rindex$ are solution indices in the left and right child search nodes' solution lists. Using these indices, ExploreCandidates recursively explores each retrieved solution to determine whether incrementing the left or right index produces new solutions with the same objective value (Line 6). Solutions matching the solution group's value are added to the solution group (see Appendix A.2), while the others are added to the candidates queue (Lines 7–8). Then, the next solution value of the branching node becomes the minimum solution value in the candidates queue (Line 9).

If the helper node is a leaf node, its next solution and candidates queue become empty (Line 11). The helper node is removed from the list of helper nodes if its candidates queue is empty or if the value of its next solution is higher than the search node's upper bound (Lines 12-13). This is because it can no longer contribute new valid solutions in future iterations. Finally, the solution group is added to the helper node's sorted solution list $SSL$ (Line 14), and it is returned as the next solution (Line 15).

Alg. 2 details the ExploreCandidates procedure. The list of solutions to explore is initialized with the same-valued solutions list (Line 2). For each solution in this list, new index pairs are generated by incrementing either the left or right index by one (Lines 6-7). If the resulting index pair has not been evaluated previously (Line 8), the left (right) child search node is queried to return its solution with the left (right) index (Lines 9-10). **If the child search node does not have that solution yet, it is obtained by calling GetNextSolution (Alg. 1).** These left and right solutions are combined into a new solution (Line 11, see Appendix A.3 for the details of the solution combination). If the combined solution has the same value as the branching node's current next solution, it is added to the same-valued solution list and is explored recursively in later iterations (Lines 13-15). Otherwise, it is added to the distinct-valued solutions list (Lines 16-18).

---

**Algorithm 1** GetNextSolution()

---

1: $node \leftarrow \arg\min_{node \in helper\_nodes} node.next$
2: $sol \leftarrow \text{CreateSolutionGroup}(node.feature, node.next)$
3: **if** $node.\text{IsBranchingNode}()$ **then**
4:     $same\_valued\_sols \leftarrow node.\text{GetSolutionsWithValue}(node.next)$
5:     $node.CQ \leftarrow node.CQ \setminus same\_valued\_sols$
6:     $same\_valued\_sols, others \leftarrow node.\text{ExploreCandidates}(same\_valued\_sols, node.next)$
7:     $sol.\text{Add}(same\_valued\_sols)$
8:     $node.CQ \leftarrow node.CQ \cup others$
9:     $node.next \leftarrow node.CQ.\text{Top}().value$
10: **else**
11:     $node.CQ \leftarrow \emptyset, \ node.next \leftarrow \infty$
12: **if** $node.CQ = \emptyset$ **or** $node.next > UB$ **then**
13:     $helper\_nodes \leftarrow helper\_nodes \setminus \{node\}$
14: $node.SSL \leftarrow node.SSL \cup sol$
15: **return** $sol$

---

---

**Algorithm 2** ExploreCandidates($same\_valued\_sols, next$)

---

1: $distinct\_valued\_sols \leftarrow \emptyset$
2: $sols\_to\_explore \leftarrow same\_valued\_sols$
3: **while** $sols\_to\_explore \neq \emptyset$ **do**
4:     $sol \leftarrow sols\_to\_explore.\text{Pop}()$
5:     **for** $left\_increment, right\_increment \in \{(1,0),(0,1)\}$ **do**
6:         $left\_index \leftarrow sol.lindex + left\_increment$
7:         $right\_index \leftarrow sol.rindex + right\_increment$
8:         **if** not $\text{Visited}(left\_index, right\_index)$ **then**
9:             $sol_L \leftarrow node_L.\text{GetNthSolution}(left\_index)$
10:             $sol_R \leftarrow node_R.\text{GetNthSolution}(right\_index)$
11:             $sol' \leftarrow \{node.feature, \text{Combine}(sol_L, sol_R)\}$
12:             $\text{Visited}(left\_index, right\_index) \leftarrow true$
13:             **if** $sol'.value = next$ **then**
14:                 $same\_valued\_sols \leftarrow same\_valued\_sols \cup sol'$
15:                 $sols\_to\_explore \leftarrow sols\_to\_explore \cup \{sol'\}$
16:             **else if** $sol'.value \leq UB$ **then**
17:                 $sol'.lindex \leftarrow left\_index, \ sol'.rindex \leftarrow right\_index$
18:                 $distinct\_valued\_sols \leftarrow distinct\_valued\_sols \cup sol'$
19: **return** $same\_valued\_sols, distinct\_valued\_sols$

---

**Depth-two subroutine** To improve scalability, Alg. 3 adapts the depth-two subroutine proposed by Demirović et al. [16] (see Sec. 3) to efficiently calculate all depth-two solutions of three branching nodes. It iterates over all pairs of features $f_i, f_j$ (Lines 1-2), with $f_i$ the branching feature in the root, and $f_j$ in either the left or right branching nodes. It then computes the optimal solutions $sol_L$ and $sol_R$ for the left and right subtree (Line 4-5). While doing so, only the solutions with values within the upper bound are kept (Lines 6-9). For each left solution, the combinations obtained with the right solutions are iteratively evaluated and inserted into the node's sorted solution list until the combination exceeds the upper bound (Lines 11-14). See Appendix A.6 for computing trees with one or two branching nodes.

If a limit is set on the Rashomon set size, computing all depth-two trees may be unnecessary. We address this by rerunning the depth-two subroutine with an incrementally increased upper bound until the actual upper bound is reached. See Appendix A.7 for details.

## 4.3 Comparison to the state-of-the-art

TreeFARMS [30] calculates the Rashomon set of trees using a depth-first search and supports only classification tasks. It requires a predefined Rashomon multiplier. Its solutions can be sorted efficiently but only after full enumeration, making correct estimation of the multiplier necessary.

**Algorithm 3** CalculateThreeNodeSols($node, F, Q^+, Q^-$)

1: **for** $f_i \in F$ **do**
2:     $left\_sols \leftarrow \emptyset, right\_sols \leftarrow \emptyset$
3:     **for** $f_j \in F, i \neq j$ **do**
4:        $sol_L \leftarrow \text{Sol}(f_j, \text{Combine}(\min\{Q^+(\bar{f_i}, f_j), Q^-(\bar{f_i}, f_j)\}, \min\{Q^+(\bar{f_i}, \bar{f_j}), Q^-(\bar{f_i}, \bar{f_j})\}))$
5:        $sol_R \leftarrow \text{Sol}(f_j, \text{Combine}(\min\{Q^+(f_i, f_j), Q^-(f_i, f_j)\}, \min\{Q^+(f_i, \bar{f_j}), Q^-(f_i, \bar{f_j})\}))$
6:        **if** $sol_L.value \leq node.UB$ **then**
7:           $left\_sols \leftarrow left\_sols \cup \{sol_L\}$
8:        **if** $sol_R.value \leq node.UB$ **then**
9:           $right\_sols \leftarrow right\_sols \cup \{sol_R\}$
10:    **for** $left \in left\_sols$ in ascending order **do**
11:       **for** $right \in right\_sols$ in ascending order **do**
12:          $val \leftarrow \text{Combine}(left, right)$
13:          **if** $val > node.UB$ **then break**
14:          $node.SSL.\text{Add}(\text{Sol}(f_i, val))$

In contrast, SORTD computes the Rashomon set in order of objective value using best-first search, enabling anytime behavior: it produces a valid Rashomon set at any point. This is particularly beneficial for high-dimensional datasets, ensuring search termination and stable performance. It additionally eliminates the need to guess the Rashomon multiplier, as the search can also stop based on a specified set size. When SORTD is run in the same way as TreeFARMS (i.e., with a fixed Rashomon bound), SORTD returns the whole Rashomon set up to two orders of magnitude faster than TreeFARMS, while using up to one order of magnitude less memory, as we will show in the next section. Furthermore, SORTD is not limited to classification tasks, supporting any separable and totally ordered objective for computation and any separable and partially ordered objective for post-evaluation.

**Limitations** Our approach is limited to binary features, which is a common limitation [e.g., 30]. And since finding even a single optimal tree is NP-hard, enumerating the Rashomon set for larger depth budgets, dataset sizes, or target set sizes becomes intractable. However, our scalability improvements extend the range of problem instances that can practically be solved.

## 5 Experimental evaluation

We conduct a series of experiments with the following aims: (1) to assess SORTD's runtime efficiency in computing Rashomon sets; (2) to showcase that a small number of high-quality trees—easily found by SORTD—may be informative for model evaluation via variable importance analysis; and (3) to demonstrate SORTD's flexibility in enumerating and analysing Rashomon sets under varying objective functions.

**Experiment set-up** For aims (1) and (2), we use the 30 benchmark binary classification datasets previously used to assess state-of-the-art methods [10, 15, 16, 30, 46]. For aim (3) we adopt common regression [47] and fairness benchmark datasets [48]. We implemented SORTD in C++ and provide it as a python package.[1] We use STreeD [40] to compute optimal trees in SORTD's first phase. All experiments are run single-threaded on an Intel Xeon E5-6448Y @ 2.1 GHz with 100 GB RAM [49], with a 300 seconds time limit. Further details are provided in Appendix B.

### 5.1 Runtime performance

We evaluated SORTD's runtime performance against the state-of-the-art method TreeFARMS [30] across varying Rashomon set sizes $n^T$. Since TreeFARMS requires the Rashomon multiplier (or bound) to be specified in advance, we ensured a fair comparison by precomputing the Rashomon multipliers (see Appendix B.2) using the following procedure. We varied the depth budget $d \in \{3, 4, 5\}$ and the complexity cost $\lambda \in \{0.001, 0.01, 0.1\}$. Using each (dataset, $d$, $\lambda$) combination, we

---
[1]`https://github.com/ConSol-Lab/pysortd`

ran SORTD to find the smallest multiplier $\varepsilon$ that yields at least $n^T = 10^n$ trees for $n \in \{1, \ldots, 6\}$. We limited $n$ to six, as we consider sets larger than $10^6$ trees to be impractical for real-world analysis. Both methods then used these multipliers for Rashomon set enumeration, and runtime was measured to obtain the *whole* Rashomon set up to the specified bound.

Fig. 3 plots the empirical cumulative distribution of runtimes for finding the whole Rashomon set of the specified size, aggregated over all datasets and $\lambda$ values (see Appendix B.2 for detailed results). Across every depth budget and Rashomon set size, SORTD consistently outperforms TreeFARMS, reaching speed-ups of up to two orders of magnitude. As the depth budget grows, SORTD's runtime increases only slightly—most sets are produced under 10 seconds even at the largest depth—whereas TreeFARMS slows by more than an order of magnitude and hits the time limit after depth budget three. Moreover, at higher depth limits, SORTD scales better for increasingly large Rashomon sets.

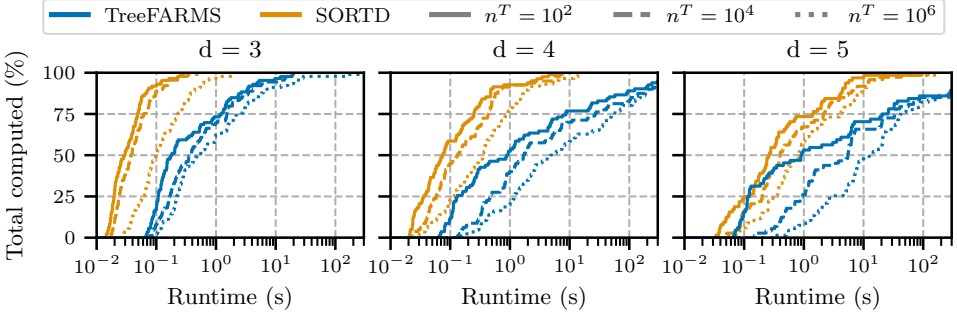

Figure 3: Cumulative runtime (s) distribution across tree depths $d$ and Rashomon set sizes $n^T$. The x-axis is logarithmic and shows the runtime for enumerating the full Rashomon set. SORTD is up to two orders of magnitude faster than TreeFARMS.

We additionally investigated how feature dimensionality affects runtime. Fig. 4 shows the empirical cumulative distribution of runtimes at depth budget four and $n^T = 10^6$ while varying the feature dimension of the input dataset. Runtime increases for both methods as dimensionality grows, but SORTD is impacted less. Once the feature count exceeds 30, TreeFARMS fails to finish computing some Rashomon sets within the time limit. In contrast, SORTD remains fast: even with 30 features, it completes all runs well below the time limit.

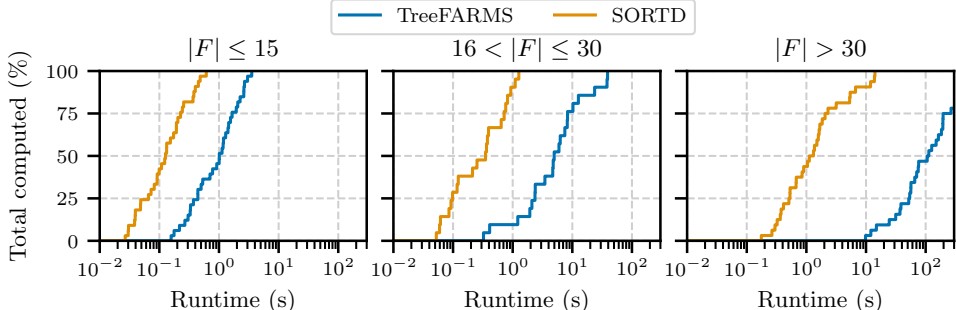

Figure 4: Cumulative runtime (s) distribution across varying feature dimensionality, with depth budget four and $n^T = 10^6$. The x-axis is logarithmic and shows the runtime for enumerating the full Rashomon set. SORTD scales better with more features than TreeFARMS.

Furthermore, we evaluated the memory usage of SORTD. Fig. 5 shows the empirical cumulative distribution of memory usage in gigabytes. For most of the instances, SORTD's memory usage remains below 1 GB, leading to on order of magnitude less memory usage. In contrast, TreeFARMS shows substantially higher memory requirements, particularly at greater depths.

## 5.2 Variable importance analysis

To test whether SORTD's in-order solution generation facilitates downstream evaluation, we measured variable importance with the *Leave-One-Feature-Out* (LOFO) score [50] by computing the increase

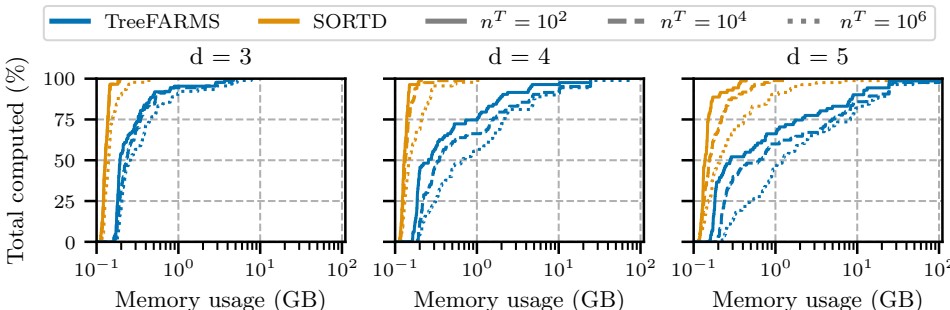

Figure 5: Cumulative memory usage (GB) distribution across tree depths $d$ and Rashomon set sizes $n^T$. Note the logarithmic x-axis. SORTD uses one order of magnitude less memory than TreeFARMS.

in the area under the Rashomon objective curve when that feature is omitted. Fig. 6 illustrates LOFO on the *compas* [51] and *fico* [52] datasets using the top-100 trees. As reported by Fisher et al. [26], "priors > 3" is an important variable in *compas*. In our analysis, omitting this feature noticeably shifts the loss curve, and a similar effect is observed when excluding external risk estimate features in *fico*, highlighting their strong predictive influence.

We now start our test whether variable importance obtained through different Rashomon set sizes provides similar insights. We treated variable importance values derived from the top-10,000 trees as a reference, and compared them with the ones obtained from the top-1 and top-100 trees. Since importance estimates within Rashomon sets can vary under resampling [25], each dataset was bootstrapped 20 times. For each resample, we computed the Rashomon multiplier required to obtain at least 10,000 trees and constructed the corresponding set using the multipliers. We then evaluated the increase in area under the Rashomon objective curve using $\lambda = 0.01$ and a depth budget of four. Due to its high runtime requirement, the *biodeg* dataset was not used in this experiment.

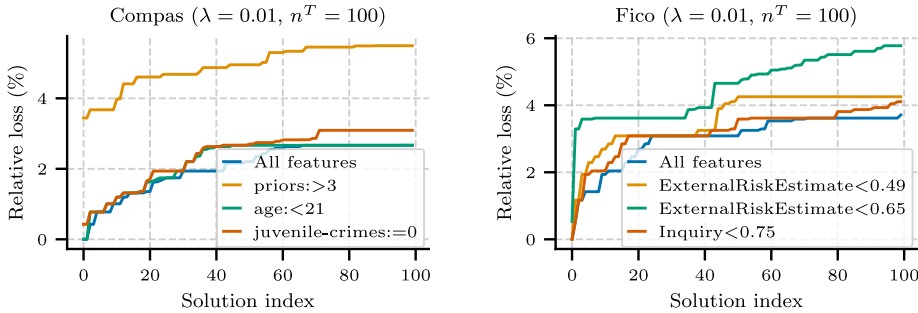

Figure 6: Top-3 influential features identified via LOFO. Shifts in the relative loss curves after removing each feature indicate their influence on the Rashomon set for the *compas* and *fico* datasets.

We evaluated top-5 feature stability using the *Jaccard index* ($|A \cap B|/|A \cup B|$) and full ranking stability using *Kendall's* $\tau$ [53]. Comparing top-1 to top-10,000 trees, 19 of 29 datasets had a Jaccard index $\geq 0.8$; this increased to 26 datasets for top-100. Similarly, $\tau \geq 0.7$ held for 5 datasets using top-1 trees, and for 17 using top-100 trees (see Appendix B.7). These results suggest that variable importance is relatively stable when using 100 trees, highlighting the potential of estimating the importance values from smaller Rashomon sets. This is particularly useful for large datasets, where computing large Rashomon sets is impractical. Developing theoretical guidelines for the minimum number of trees required to obtain stable importance estimates could further facilitate this analysis.

### 5.3 Evaluating other objectives

SORTD also supports objectives beyond accuracy (see Appendix A.9), optimizing separable totally ordered objectives directly and evaluating separable and partially ordered ones indirectly.

**Regression**  For example, SORTD can directly find the Rashomon set of sparse regression trees with the totally ordered objective of minimal mean-squared error. We demonstrate this capability by comparing SORTD with the heuristic CART [2] and the optimal method STreeD [40] since no other method is known for generating this set. Both methods repeatedly compute trees for random samples of the data until a given time limit. Fig. 7 shows that SORTD finds trees in order until the 10% relative bound is reached, whereas, as observed in [30] for classification tasks, CART and STreeD find orders of magnitude fewer trees in the Rashomon set. See Appendix B.8 for further details.

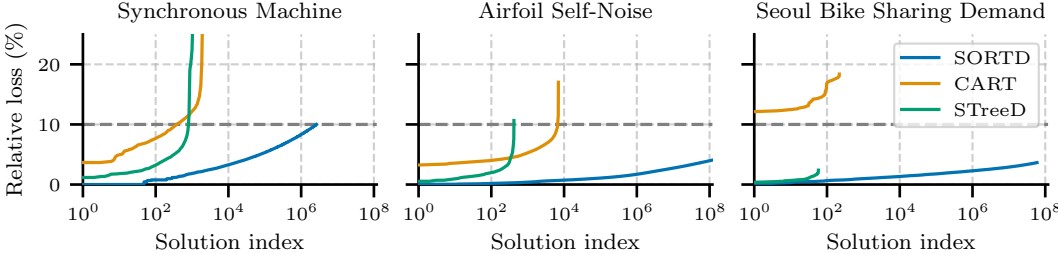

Figure 7: Relative loss compared to the optimal solution of all regression trees found within the one minute time-out for max-depth four, $\lambda = 0.001$, and $\varepsilon = 0.1$. Dashed lines indicate the Rashomon bound. SORTD finds orders of magnitude more trees in the Rashomon set than the other methods.

**Equality of opportunity**  SORTD can also evaluate partially ordered objectives such as multi-objective criteria indirectly. E.g., to find fair accurate trees, SORTD can first obtain the Rashomon set based on accuracy, and then evaluate it using another objective such as *equality-of-opportunity*, i.e., the difference in the true positive rate of two discrimination-sensitive groups. For example, Fig. 8 shows the top $10^7$ trees in the Rashomon set for accuracy evaluated with the equality-of-opportunity metric. In Appendix B.9 we provide further details and a runtime comparison with STreeD.

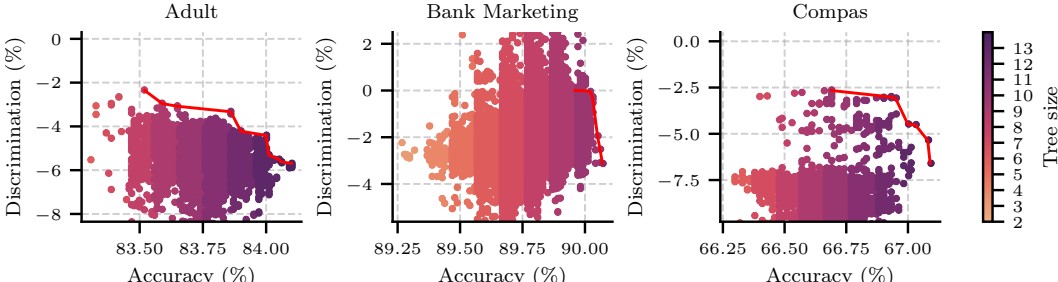

Figure 8: The top $10^7$ trees in the accuracy Rashomon set evaluated using *equality-of-opportunity* with max-depth four and $\lambda = 0.001$. The red lines show the Pareto-front of accuracy and fairness. The colours indicate tree size. The sign of the discrimination shows which group is disadvantaged.

## 6  Conclusion

The Rashomon set of sparse decision trees offers many advantages over relying on a single model, provided it can be explored efficiently. SORTD makes such exploration practical: it enumerates trees in ascending objective order, allowing the highest-quality candidates to be retrieved and evaluated quickly. At the same time, its algorithmic design scales significantly better than the state-of-the-art and uses less memory as either the feature dimensionality or the depth budget grows. Finally, it supports separable totally ordered objectives for Rashomon set computation, as well as separable and partially ordered objectives for fast post-hoc evaluation. Together, these advances turn large-scale Rashomon-set analysis and model selection into a viable option for real-world applications.

## 7  Acknowledgments

This research was supported by the TU Delft AI Labs program.

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

# A Detailed method description

## A.1 Main algorithm of Rashomon set calculation

We generate the Rashomon set by iteratively computing trees in ascending order of their objective values. Alg. 4 summarizes how the Rashomon set is computed once the optimal solution has been found. The algorithm takes as input: the maximum depth $d$, the training dataset $D$, the optimal solution value $sol^*$, the Rashomon multiplier $\varepsilon$, the maximum number of trees $n^T$, and a cache of optimal solutions of the subtrees. The Rashomon multiplier and the maximum number of trees are complementary: specifying either one is sufficient. If the Rashomon multiplier is omitted, a large default value is used to derive the bound.

Next, the root search node is initialized by setting its upper bound to the Rashomon bound and retrieving its optimal solution from the cache (Line 2). This optimal value is then compared with the bound to determine whether any solutions lie within the Rashomon set. If the condition is satisfied, additional solutions are iteratively added to the node's sorted solution list until a stopping criterion is met (Lines 4–6).

---

**Algorithm 4** $\mathrm{Main}(d, D, sol^*, \varepsilon, n^T, cache)$

---

1: $\theta \leftarrow sol^*.value \times (1 + \varepsilon)$
2: $root\_search\_node \leftarrow \mathrm{InitializeSearchNode}(d, D, \theta, cache)$
3: $sol \leftarrow root\_search\_node.sol^*$
4: **while** $sol.value \leq \theta$ **and** $|root\_search\_node.SSL| < n^T$ **do**
5: $\quad sol \leftarrow root\_search\_node.\mathrm{GetNextSolution}()$
6: $\quad root\_search\_node.SSL \leftarrow root\_search\_node.SSL \cup \{sol\}$
7: **return** $root\_search\_node.SSL$

---

## A.2 Solutions structure

A Rashomon set may contain well over a billion trees, and enumerating such a large set has a high computational and memory load. Therefore, we adopt the grouped solution structure used in [30]: Fig. 9 shows how (partial) solutions (i.e., subtrees) are stored in memory. Solutions are recursively grouped by their objective value and the splitting feature of the (subtree) root node, followed by a list of pairs of solutions for the left and right subtrees. The details of how same-valued solutions are grouped are given in Sec. 4.2.

**Example 3.** *In Fig. 9, the left of the figure represents different solution values of a branching node. Because the first two solution values are the same, a solution group (right upper side of the image) with two solution pairs is created. As for the remaining solution value, a solution group (right lower side of the image) with one solution pair is created.*

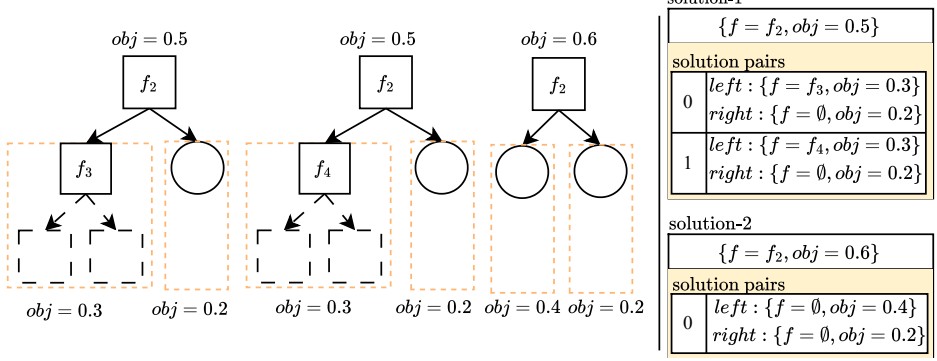

Figure 9: SORTD grouped solution structure.

## A.3 Incorporating the regularization cost

The regularization term in the objective function penalizes model complexity by adding $\lambda$ for every leaf node (Eq. (1)). Consequently, creating a new split that raises the leaf node count by one increases the objective value even if predictive accuracy is unchanged. To capture this effect, we assign a fixed cost of $\lambda$ to each branching decision. If the tree consists of a single root leaf, the same cost $\lambda$ is added once for that leaf node.

We use the branching cost while combining left and right solutions and while determining the upper bounds of the child search nodes (see Appendix A.4).

To construct a solution for a branching (search) node, the algorithm first calculates solutions for its left and right child nodes. It then combines these partial solutions to obtain the parent solution (Alg. 2, Line 11). Since the learning objective imposes a regularization penalty of $\lambda$ per additional leaf, we add the same branching cost of $\lambda$ when combining the left and right subtree solutions as follows:

$$\text{Combine}(sol_L, sol_R) = sol_L.value + sol_R.value + \lambda. \tag{3}$$

## A.4 Upper bounding

To avoid exploring low–quality regions that cannot contribute to the Rashomon set, every search node is equipped with an upper bound on the solution value of any tree it may still yield. The root node's upper bound is initialized with the Rashomon bound, and each helper node simply inherits its parent's upper bound. Child search nodes, however, receive tighter bounds that account for the best solution already attainable in the complementary subtree. Given a branching node and its upper bound $UB$, the upper bounds of the child search nodes are calculated as follows:

$$\text{UB}_\text{L}(UB, sol_R^*) = UB - sol_R^* - \lambda, \tag{4}$$
$$\text{UB}_\text{R}(UB, sol_L^*) = UB - sol_L^* - \lambda. \tag{5}$$

Here, $\lambda$ is to account for the branching cost (see Appendix A.3) while $sol_L^*$ and $sol_R^*$ are the optimal solution values of the left and right child search nodes, respectively. These optimal solutions are either retrieved from a cache of optimal solutions or computed using STreeD [40] if absent.

We also use the upper bounds of branching and search nodes for two additional purposes: (1) to filter out helper nodes whose next solution value exceeds its search node's upper bound (see Alg. 1); and (2) to determine whether a newly generated solution obtained by combining a left and a right solution should be accepted, by comparing it against the branching node's upper bound (see Alg. 2).

## A.5 Caching

A subtree can be encountered multiple times during the search. To avoid recomputing its solutions in different parts of the search space, we cache both solutions themselves—stored in the search node's sorted solution list—and the objects required to compute them, namely the branching search nodes.

We adopt the caching mechanism of Demirović et al. [16], which supports two strategies: *dataset caching*, in which a subtree is identified by the set of samples it contains, and *branch caching*, in which it is identified by the features used in every branching decision from the root to that subtree. In our experiments, we used dataset caching.

When a new search node is created, the algorithm first queries the cache. If the subtree is present, the node reuses the cached solution list and branching nodes. The solution list is then extended—if not already exhausted—by all search nodes that share it.

## A.6 Depth-two subroutine

We use a specialized algorithm to calculate the Rashomon set when the remaining depth budget is at most two. Sec. 4.2 details how, for a given search node, we enumerate all trees that contain three branching nodes. Below, we describe the cases with one and two branching nodes.

Alg. 5 outlines the procedure for generating all trees with a single branching node. The algorithm takes as input a search node $node$, a feature set $F$, class frequency counts $Q^+$ and $Q^-$, and an upper bound $UB$. For each feature in $F$, the algorithm computes the label assignments for the left and right

children that minimize their respective subtree costs. These two partial solutions are then combined (Line 2; see Appendix A.3 for details). If the resulting solution value is within the upper bound, the solution is added to the search node's sorted solution list (Lines 3-4).

---

**Algorithm 5** CalculateOneNodeSol($node, F, Q^+, Q^-, UB$)

---

1: **for** $f_i \in F$ **do**
2:     $val \leftarrow \text{Combine}(\min\{Q^+(\bar{f_i}), Q^-(\bar{f_i})\}, \min\{Q^+(f_i), Q^-(f_i)\})$
3:     **if** $val \leq UB$ **then**
4:        $node.SSL.\text{SortedInsert}(\text{Sol}(f_i, val))$

---

For trees with exactly two branching nodes, the feasible topology is restricted to one of two mirror images: either the left subtree is a leaf and the right subtree contains a single branching node, or vice versa. Alg. 6 handles the case in which the right child is a leaf; the symmetric case is treated analogously.

The procedure begins by selecting a feature for the root split and computing the solution of the right subtree being a leaf (Line 3). Next, it creates a second split in the left subtree using any remaining feature from $F$ and evaluates every possible label assignment, selecting the minimal left-subtree solution value (Line 5). If this value does not exceed the upper bound, the left-subtree solution is stored (Lines 6-7). Each stored left solution is then combined with the right-leaf solution. The resulting trees are inserted into the search node's sorted solution list whenever its overall solution value remains within the upper bound (Lines 8-12).

---

**Algorithm 6** CalculateTwoNodeSols($node, F, Q^+, Q^-, UB$)

---

1: **for** $f_i \in F$ **do**
2:     $left\_sols \leftarrow \emptyset$
3:     $val_R \leftarrow \min\{Q^+(f_i), Q^-(f_i)\}$
4:     **for** $f_j \in F, i \neq j$ **do**
5:        $val_L = \text{Combine}(\min\{Q^+(\bar{f_i}, f_j), Q^-(\bar{f_i}, f_j)\}, \min\{Q^+(\bar{f_i}, \bar{f_j}), Q^-(\bar{f_i}, \bar{f_j})\})$
6:        **if** $val_L \leq UB$ **then**
7:           $left\_sols \leftarrow left\_sols \cup \text{Sol}(f_j, val_L)$
8:     **for** $left \in left\_sols$ **do**
9:        $val \leftarrow \text{Combine}(left.value, val_R)$
10:       **if** $val \leq UB$ **then**
11:          $sol \leftarrow \text{UpdateSolution}(f_i, val)$
12:          $node.SSL.\text{SortedInsert}(sol)$
       *\*\*The case where the left tree is a leaf node and the right tree has a branching node is calculated analogously.\*\**

---

### A.7   Gradual solution creation

When the Rashomon set is subject to a size limit (see Alg. 4), enumerating only a subset of the depth-two trees can suffice. We achieve this by tightening the upper bound, hence pruning more candidates, reducing memory usage and runtime. If the bound is too low, however, the algorithm may spend significant effort (e.g., computing frequency counts) only to yield a single solution; if it is too high, many unused trees are still generated. Empirically, a bound that guarantees the inclusion of the leaf solution and all trees with at most one branching node strikes a good balance. Therefore, we initialize the depth-two upper bound as

$$\hat{UB} \; = \; \min\{\, \text{C}(T, D) + \lambda, \; UB \,\},  \tag{6}$$

where $D$ and $UB$ denote the dataset and the search node's bound, $T$ is the single-leaf tree at that node, and $\lambda$ is the branching cost (Appendix A.3).

With $\hat{UB}$ set, Alg. 3, 5, and 6 generate the depth-two solutions. If the calculated solutions are exhausted before the Rashomon-set size limit is reached, we relax the bound as follows:

$$\hat{UB} \leftarrow \begin{cases} UB & (UB - \hat{UB}) < 0.2 \cdot UB \\ \hat{UB} + 0.5\,(UB - \hat{UB}) & \text{otherwise} \end{cases}  \tag{7}$$

This increases the upper bound at a diminishing rate, as the number of solutions grows exponentially with respect to the changes in the upper bound.

## A.8 Ignoring trivial extensions

By default, SORTD allows extending a tree node with two child nodes that share the same label. Following Xin et al. [30], we refer to these as *trivial extensions*. SORTD can optionally disregard such trees when calculating the Rashomon set. A trivial extension occurs when a branching node produces two leaf nodes whose labels are identical, so the branching does not change the objective value except for the added complexity cost. A trivial extension is detected during the depth-two subroutine (e.g., Alg. 3 Line 12) or branch-tracker iterations (Alg. 2 Line 11) if the following condition holds:

$$\text{IsLeaf}(left) \text{ and } \text{IsLeaf}(right) \text{ and } left.label = right.label \tag{8}$$

In some cases, at least one leaf label may be arbitrary: assigning one label or another does not affect the objective value. In this case, SORTD assigns one label arbitrarily, stores one of the alternatives, and when combining with another leaf node, selects the label that avoids creating a trivial extension.

Removing trivial extensions reduces the number of trees in a Rashomon set for a fixed Rashomon multiplier, or equivalently, increases the Rashomon multiplier attainable for a fixed set size. As shown in Appendix B.6, SORTD maintains performance improvements of up to two orders of magnitude also when trivial extensions are excluded.

## A.9 Optimizing other objectives

SORTD supports computing Rashomon sets for separable and totally ordered objectives *directly* and post-evaluating other separable and partially-ordered objectives over an already computed Rashomon set *indirectly*.

**Optimizing totally ordered objectives** SORTD's Rashomon construction is based on the STreeD framework [40], and therefore easily generalizes to other objectives than the regularized misclassification score. However, since SORTD requires sorting partial solutions in its search nodes, it cannot support all optimization tasks that STreeD supports i.e., all separable and partially ordered tasks. Specifically, SORTD requires *separable and totally ordered* solutions. Additionally, in our implementation, we assume all objectives are *additive*, i.e., solutions from left and right subtrees can be combined using addition. SORTD therefore, has the same generalizability as DL8 [54].

To support other objectives, the only major change is to redefine the objective function $C(T, D)$. For example, to optimize regression trees by minimizing the *sum of squared errors*, we define:

$$C(T, D) = \sum_{(x,y) \in D} (T(x) - y)^2 + \lambda N(T). \tag{9}$$

To illustrate this, we show in Appendix B.8 SORTD's performance in finding the Rashomon set for regression trees.

**Post-evaluating other objectives** After generating a Rashomon set, the trees in this set can be evaluated using any objective by going over each tree one at a time. However, computing an objective for each tree individually can be expensive. Therefore, we provide a special procedure to post-evaluate objectives that are *separable* (as defined by Linden et al. [40]), i.e., we can compute the objective values separately for the left and right subtrees and combine them afterwards. This requires defining some cost function $g(D, k)$ for leaf nodes with $D$ the input dataset and $k$ the assigned label. This cost function does not necessarily need to return a real-valued number, but can also, for example, return a tuple of values. Similarly, we need to define a combining operator $\oplus$ that takes a left and right solution and returns the cost of combining both (again, not necessarily a real-valued number).

SORTD uses these functions to construct a new Rashomon set, with equal-valued solutions grouped together (see Appendix A.2). However, in this post-evaluation, we no longer require the objective to be totally ordered, so we can no longer acquire the solutions in order of the second objective. In practice, we obtain the Rashomon set in order according to the optimization objective (e.g., regularized accuracy), by retrieving them in batches. We run Alg. 4 for a certain Rashomon set size; evaluate the obtained solutions using the second objective; and possibly rerun Alg. 4 with a larger

Rashomon set size, continuing where we left. This can be repeated until a certain time-out or some other user-defined stopping criterion.

To facilitate ease of evaluation, SORTD allows users to define the leaf cost function $g$ and the combining operator $\oplus$ in Python and pass them to SORTD. Consider, for example, optimizing trees with a multi-objective criterion of regularized accuracy and the fairness metric *equality-of-opportunity*. The equality of opportunity loss is defined as the difference in the true positive rate between two discrimination sensitive groups. More formally, let $y$ be the true label, $\hat{y}$ the predicted label, then equality of opportunity requires:

$$P(\hat{y} = 1 \mid y = 1, \text{group one}) = P(\hat{y} = 1 \mid y = 1, \text{group two}). \tag{10}$$

We can compute this by defining a leaf cost function that counts for both groups how many samples with the true positive label are also predicted with a positive label. Additionally, we count the total number of misclassifications, so that we can also compute the accuracy. Hence, the solution tuple returned will be $(misclassifications, positive\_count\_group\_0, positive\_count\_group\_1)$. The combining operator is then simply element-wise addition of two tuples. Finally, we need to provide a function that computes from these tuples in the root node the actual objective value. In this example, the final objective value is a tuple of the accuracy and the difference in the true positive rate for the two groups. These tuples can then be filtered to only keep the Pareto front. In Python, we can write:

```python
def leaf(dataview, label):
  mis = dataview.num_instances_for_label(1 - label)
  if label == 1:
    # Assume feature 0 is the discrimination sensitive feature
    group1_size = dataview.num_instances_for_label_and_feature(1, 0)
    group0_size = dataview.num_instances_for_label(1) - group1_size
    return (mis, group0_size, group1_size)
  return (mis, 0, 0)

def add(sol1: tuple, sol2: tuple):
  return (sol1[0] + sol2[0], sol1[1] + sol2[1], sol1[2] + sol2[2])

def obj(sol: tuple):
  acc  = 1.0 - sol[0] / N
  disc = (sol[1] / N_pos_group0) - (sol[2] / N_pos_group1)
  return (acc, disc)

model = SORTDClassifier("cost-complex-accuracy", max_depth=3,
    cost_complexity=0.01, max_num_trees = 10000)
model.fit(X, y)
results = model.evaluate_other_objective(X, y, leaf, add)
solution_values = [obj(s.objective) for s in results]
```

Listing 1: Example Python Code for optimizing accuracy and equality of opportunity.

# B   Experiment details

## B.1   Classification datasets

We selected all 46 binary classification datasets with less than 100 binary features from [16] and [30].[2] We excluded nine duplicate datasets and three trivially solvable ones (solved in under one second). We also removed four datasets that exceeded memory limits when calculating the Rashomon multipliers for specific Rashomon set sizes. This computation was memory intensive because the Rashomon multipliers were unknown, which required setting the upper bound to a high value.

After these exclusions, 30 datasets remained. From these, we eliminated duplicate and complementary features. Table 1 lists the datasets together with their sample size and resulting feature size, $|D|$ and $|F|$, respectively. These datasets were used in runtime (Sec. 5.1) and variable importance analysis (Sec. 5.2). The original datasets can be obtained from the UCI Machine Learning repository [55] and from [51, 52, 56, 57].

---

[2]https://bitbucket.org/EmirD/murtree, https://github.com/ubc-systopia/treeFarms

Table 1: Classification datasets

| Dataset | $|D|$ | $|F|$ | Dataset | $|D|$ | $|F|$ |
|---|---|---|---|---|---|
| anneal | 812 | 44 | HTRU2 | 17898 | 57 |
| bank | 4521 | 23 | hypothyroid | 3247 | 39 |
| banknote | 1372 | 16 | kr-vs-kp | 3196 | 38 |
| bar-7 | 1913 | 14 | lymph | 148 | 47 |
| biodeg | 1055 | 81 | messidor | 1151 | 24 |
| breast | 699 | 10 | monk1 | 124 | 15 |
| car | 1728 | 15 | monk2 | 169 | 11 |
| cheap | 2653 | 15 | monk3 | 122 | 15 |
| coffee | 3816 | 15 | mouse | 70 | 45 |
| compas | 6907 | 12 | soybean | 630 | 42 |
| diabetes | 768 | 11 | spect | 267 | 22 |
| expensive | 1417 | 15 | tic-tac-toe | 958 | 18 |
| fico | 10459 | 17 | tumor | 336 | 17 |
| haberman | 306 | 92 | vote | 435 | 48 |
| hepatitis | 137 | 34 | yeast | 1484 | 46 |

## B.2 Runtime performance

We generated benchmark instances by varying the complexity penalty $\lambda \in \{0.001, 0.01, 0.1\}$ and the depth budget $d \in \{3, 4, 5\}$. For each (dataset, $\lambda$, $d$) combination, we use SORTD to compute the minimum Rashomon multiplier (and corresponding Rashomon bound) that yields at least $10^n$ trees for every $n \in \{1, \ldots, 6\}$. Because both methods employ a grouped solution structure, requesting $10^n$ solutions may sometimes yield more than $10^{n+1}$ solutions, with the additional solutions sharing the same objective value. In such cases, we did not generate a separate instance for the $10^{n+1}$ target, as it was already satisfied. Finally, we ran each instance five times.

We compare SORTD with the state-of-the-art algorithm TreeFARMS [30], under a 300-second time limit per instance. Each experiment is repeated five times. We set `rashomon_ignore_trivial_extensions = False` for both methods (see Sec. B.6 for runtime results with trivial extensions ignored). Runtime results are presented in Tables 2-4. The results of the instances with depth budget three are not presented as both methods solved most of them within a second. Entries marked '–' indicate omitted runs due to excessive memory usage. We denote runtimes below one second with '$< 1$', and timeouts with '$> 300$'.

As expected, increasing $\lambda$ simplifies the search problem for both methods. A larger penalty on leaf usage encourages shallower trees, which effectively prunes deeper parts of the search space. In contrast, small $\lambda$ values lead to significantly larger search spaces.

Across most instances, SORTD completes enumeration within one second for depth four and within ten seconds for depth five. In contrast, TreeFARMS exhibits high variability—even for depth four, runtimes range from below a second to full timeouts. Its performance degrades further with increasing depth, frequently timing out for $\lambda = 0.001$ and $\lambda = 0.01$ especially when the feature size of the dataset is high.

To evaluate the overall performance of SORTD compared to TreeFARMS, Tables 2-4 also report the geometric mean of the average runtime ratios $t_{\text{TreeFARMS}}/t_{\text{SORTD}}$, where timeout values are considered as 300 seconds. Results show that SORTD achieves up to two orders of magnitude improvement in runtime over TreeFARMS for Rashomon set enumeration. These findings further emphasize SORTD's scalability, particularly as problem complexity increases with respect to tree depth, dataset size, and regularization parameter $\lambda$.

Table 2: Runtime (s) performance of methods to calculate Rashomon sets across datasets and depth budgets with $\lambda = 0.001$ and $n^T = 10^n$, $n \in \{1,..,6\}$. Results are reported as mean $\pm$ standard error. A '−' indicates results that are omitted due to their high memory requirement.

| Dataset | $|D|$ | $|F|$ | $d = 4$ TreeFARMS | $d = 4$ SORTD | $d = 5$ TreeFARMS | $d = 5$ SORTD |
|---|---|---|---|---|---|---|
| breast | 699 | 10 | <1 | <1 | <1 | <1 |
| monk2 | 169 | 11 | <1 | <1 | 2±0.1 | <1 |
| diabetes | 768 | 11 | <1 | <1 | <1 | <1 |
| compas | 6907 | 12 | <1 | <1 | 1±0.0 | <1 |
| bar-7 | 1913 | 14 | <1 | <1 | 1±0.0 | <1 |
| car | 1728 | 15 | <1 | <1 | 5±0.2 | <1 |
| expensive | 1417 | 15 | <1 | <1 | 6±0.1 | <1 |
| cheap | 2653 | 15 | 1±0.0 | <1 | 6±0.1 | <1 |
| monk1 | 124 | 15 | <1 | <1 | 4±0.7 | <1 |
| monk3 | 122 | 15 | 1±0.0 | <1 | 22±0.7 | <1 |
| coffee | 3816 | 15 | 1±0.0 | <1 | 7±0.2 | <1 |
| banknote | 1372 | 16 | <1 | <1 | <1 | <1 |
| tumor | 336 | 17 | 2±0.1 | <1 | 20±0.8 | <1 |
| fico | 10459 | 17 | 5±0.2 | <1 | 32±1.1 | 2±0.0 |
| tic-tac-toe | 958 | 18 | 2±0.0 | <1 | 21±0.0 | <1 |
| spect | 267 | 22 | 22±0.7 | <1 | >300 | <1 |
| bank | 4521 | 23 | 8±0.1 | <1 | 88±0.4 | 2±0.0 |
| messidor | 1151 | 24 | 5±0.3 | <1 | 33±0.7 | <1 |
| hepatitis | 137 | 34 | 103±2.6 | <1 | >300 | 61±14.8 |
| kr-vs-kp | 3196 | 38 | 46±0.6 | <1 | >300 | 12±0.1 |
| hypothyroid | 3247 | 39 | 49±0.8 | <1 | >300 | 9±0.5 |
| soybean | 630 | 42 | 57±2.0 | <1 | >300 | 7±0.2 |
| anneal | 812 | 44 | 24±0.3 | <1 | >300 | 5±0.1 |
| mouse | 70 | 45 | 9±0.1 | <1 | >300 | 5±0.1 |
| yeast | 1484 | 46 | 184±11.3 | <1 | >300 | 14±0.4 |
| lymph | 148 | 47 | 139±6.1 | <1 | >300 | 9±0.0 |
| vote | 435 | 48 | >300 | <1 | >300 | 13±0.1 |
| HTRU2 | 17898 | 57 | >300 | 5±0.2 | >300 | 140±2.5 |
| biodeg | 1055 | 81 | >300 | 4±0.0 | >300 | 203±5.9 |
| haberman | 306 | 92 | >300 | 2±0.0 | >300 | 155±3.8 |
| **Geometric mean improvement** | | | | 86.95 | | 24.30 |

Table 3: Runtime (s) performance of methods to calculate Rashomon sets across datasets with $\lambda = 0.01$ and $n^T = 10^n$, $n \in \{1,..,6\}$. Results are reported as mean $\pm$ standard error. A '−' indicates results that are omitted due to their high memory requirement.

| Dataset | $|D|$ | $|F|$ | $d = 4$ TreeFARMS | $d = 4$ SORTD | $d = 5$ TreeFARMS | $d = 5$ SORTD |
|---|---|---|---|---|---|---|
| breast | 699 | 10 | <1 | <1 | <1 | <1 |
| monk2 | 169 | 11 | <1 | <1 | 2±0.1 | <1 |
| diabetes | 768 | 11 | <1 | <1 | <1 | <1 |
| compas | 6907 | 12 | <1 | <1 | <1 | <1 |
| bar-7 | 1913 | 14 | <1 | <1 | 1±0.0 | <1 |
| car | 1728 | 15 | <1 | <1 | 5±0.2 | <1 |
| expensive | 1417 | 15 | <1 | <1 | 6±0.1 | <1 |
| cheap | 2653 | 15 | 1±0.1 | <1 | 5±0.4 | <1 |
| monk1 | 124 | 15 | <1 | <1 | 5±1.1 | <1 |
| monk3 | 122 | 15 | 2±0.1 | <1 | 32±1.5 | <1 |
| coffee | 3816 | 15 | 1±0.1 | <1 | 7±0.3 | <1 |

| | | | $d = 4$ | | $d = 5$ | |
|---|---|---|---|---|---|---|
| Dataset | $|D|$ | $|F|$ | TreeFARMS | SORTD | TreeFARMS | SORTD |
| banknote | 1372 | 16 | <1 | <1 | 1±0.0 | <1 |
| tumor | 336 | 17 | 2±0.1 | <1 | 19±0.5 | <1 |
| fico | 10459 | 17 | 5±0.2 | <1 | 37±1.6 | 3±0.1 |
| tic-tac-toe | 958 | 18 | 2±0.0 | <1 | 21±0.1 | <1 |
| spect | 267 | 22 | 33±1.2 | <1 | >300 | <1 |
| bank | 4521 | 23 | 7±0.4 | <1 | 49±5.8 | 3±0.2 |
| messidor | 1151 | 24 | 5±0.4 | <1 | 33±2.8 | <1 |
| hepatitis | 137 | 34 | 125±2.4 | <1 | >300 | 72±9.2 |
| kr-vs-kp | 3196 | 38 | 46±0.9 | <1 | >300 | 11±0.2 |
| hypothyroid | 3247 | 39 | 21±3.5 | <1 | 79±20.3 | 6±0.4 |
| soybean | 630 | 42 | 102±10.0 | <1 | >300 | 5±0.2 |
| anneal | 812 | 44 | 25±0.7 | <1 | >300 | 5±0.2 |
| mouse | 70 | 45 | 11±0.3 | <1 | >300 | 5±0.1 |
| yeast | 1484 | 46 | 189±10.7 | <1 | >300 | 13±0.7 |
| lymph | 148 | 47 | 241±13.9 | <1 | >300 | 6±0.1 |
| vote | 435 | 48 | >300 | <1 | >300 | 8±0.5 |
| HTRU2 | 17898 | 57 | 198±24.7 | 8±0.4 | 224±24.9 | 117±6.0 |
| biodeg | 1055 | 81 | >300 | 4±0.1 | - | - |
| haberman | 306 | 92 | >300 | 3±0.3 | >300 | 133±2.7 |
| **Geometric mean improvement** | | | | 68.57 | | 24.60 |

Table 4: Runtime (s) performance of methods to calculate Rashomon sets across datasets with $\lambda = 0.1$ and $n^T = 10^n$, $n \in \{1, .., 6\}$. Results are reported as mean ± standard error.

| | | | $d = 4$ | | $d = 5$ | |
|---|---|---|---|---|---|---|
| Dataset | $|D|$ | $|F|$ | TreeFARMS | SORTD | TreeFARMS | SORTD |
| breast | 699 | 10 | <1 | <1 | <1 | <1 |
| monk2 | 169 | 11 | <1 | <1 | 1±0.3 | <1 |
| diabetes | 768 | 11 | <1 | <1 | <1 | <1 |
| compas | 6907 | 12 | <1 | <1 | <1 | <1 |
| bar-7 | 1913 | 14 | <1 | <1 | <1 | <1 |
| car | 1728 | 15 | <1 | <1 | 2±0.5 | <1 |
| expensive | 1417 | 15 | <1 | <1 | 2±0.5 | <1 |
| cheap | 2653 | 15 | <1 | <1 | 1±0.3 | <1 |
| monk1 | 124 | 15 | 1±0.2 | <1 | 11±2.7 | <1 |
| monk3 | 122 | 15 | <1 | <1 | 4±1.3 | <1 |
| coffee | 3816 | 15 | 1±0.2 | <1 | 2±0.5 | <1 |
| banknote | 1372 | 16 | <1 | <1 | <1 | <1 |
| tumor | 336 | 17 | 1±0.2 | <1 | 2±0.7 | <1 |
| fico | 10459 | 17 | 2±0.6 | <1 | 4±1.2 | 2±0.3 |
| tic-tac-toe | 958 | 18 | 2±0.3 | <1 | 6±1.4 | <1 |
| spect | 267 | 22 | 7±2.8 | <1 | 11±4.3 | <1 |
| bank | 4521 | 23 | 3±0.8 | <1 | 4±1.4 | 2±0.3 |
| messidor | 1151 | 24 | 2±0.5 | <1 | 2±0.7 | <1 |
| hepatitis | 137 | 34 | 14±5.1 | <1 | 15±5.9 | 12±2.4 |
| kr-vs-kp | 3196 | 38 | 32±4.7 | <1 | 125±25.2 | 4±0.6 |
| hypothyroid | 3247 | 39 | 4±1.0 | <1 | 5±1.4 | 5±0.8 |
| soybean | 630 | 42 | 15±6.0 | <1 | 22±9.0 | 4±0.7 |
| anneal | 812 | 44 | 3±0.7 | <1 | 3±1.0 | 3±0.5 |
| mouse | 70 | 45 | 9±2.5 | <1 | 56±21.8 | 1±0.2 |
| yeast | 1484 | 46 | 66±13.3 | 1±0.2 | 131±24.5 | 10±1.5 |
| lymph | 148 | 47 | 50±14.5 | <1 | 73±20.5 | 3±0.5 |
| vote | 435 | 48 | 10±2.9 | <1 | 10±3.0 | 4±0.8 |

| Dataset | $|D|$ | $|F|$ | $d = 4$ | | $d = 5$ | |
|---|---|---|---|---|---|---|
| | | | TreeFARMS | SORTD | TreeFARMS | SORTD |
| HTRU2 | 17898 | 57 | 30±9.7 | 6±0.8 | 35±11.6 | 51±11.5 |
| biodeg | 1055 | 81 | 170±27.3 | 6±0.8 | - | - |
| haberman | 306 | 92 | 255±21.1 | 2±0.3 | 265±22.5 | 21±3.7 |
| **Geometric mean improvement** | | | | 16.06 | | 6.10 |

**Longer runtime evaluation for timeout instances** Runtime results in Tables 2-4 show that several TreeFARMS [30] instances reached the 300-second time limit. To further examine these cases, we extended the runtime limit to 3600 seconds. Tables 5-6 present the results for depth budgets four and five, respectively.

With the extended limit, most depth-four instances complete within the allotted time. For depth five, TreeFARMS requires substantially more memory for some larger instances and longer runtimes for others, highlighting SORTD's improvements in runtime and memory efficiency.

Table 5: Runtime (s) of methods for calculating the Rashomon set of the timeout instances in Sec. B.2, with a depth budget of four, aggregated for each dataset and $\lambda$.

| Dataset | $\lambda$ | TreeFARMS | SORTD |
|---|---|---|---|
| biodeg | 0.001 | >3600 | 4 |
| biodeg | 0.01 | >3600 | 4 |
| biodeg | 0.1 | 1499 | 9 |
| haberman | 0.001 | 653 | 2 |
| haberman | 0.01 | 2025 | 3 |
| haberman | 0.1 | 387 | 3 |
| HTRU2 | 0.001 | 754 | 5 |
| HTRU2 | 0.01 | 641 | 10 |
| lymph | 0.01 | - | <1 |
| lymph | 0.1 | - | 2 |
| vote | 0.001 | 454 | <1 |
| vote | 0.01 | 406 | <1 |
| vote | 0.1 | - | 2 |
| **Geometric mean improvement** | | | 325.64 |

Table 6: Runtime (s) of methods for calculating the Rashomon set of the timeout instances in Sec. B.2, with a depth budget of five, aggregated for each dataset and $\lambda$. A '−' indicates results omitted due to high memory requirements.

| Dataset | $\lambda$ | TreeFARMS | SORTD |
|---|---|---|---|
| anneal | 0.001 | 570 | 6 |
| anneal | 0.01 | 513 | 6 |
| biodeg | 0.001 | - | 195 |
| biodeg | 0.01 | - | 104 |
| haberman | 0.001 | >3600 | 154 |
| haberman | 0.01 | - | 145 |
| haberman | 0.1 | - | 27 |
| hepatitis | 0.001 | 2318 | 57 |
| hepatitis | 0.01 | >3600 | 80 |
| HTRU2 | 0.001 | - | 134 |
| HTRU2 | 0.01 | - | 125 |
| HTRU2 | 0.1 | - | 170 |
| hypothyroid | 0.001 | 1654 | 9 |
| hypothyroid | 0.01 | - | 9 |

| Dataset | $\lambda$ | TreeFARMS | SORTD |
|---|---|---|---|
| kr-vs-kp | 0.001 | 1250 | 12 |
| kr-vs-kp | 0.01 | 1323 | 11 |
| kr-vs-kp | 0.1 | 369 | 7 |
| lymph | 0.001 | 1888 | 9 |
| lymph | 0.01 | >3600 | 6 |
| lymph | 0.1 | - | 6 |
| mouse | 0.001 | 1195 | 4 |
| mouse | 0.01 | 1031 | 5 |
| mouse | 0.1 | - | 3 |
| soybean | 0.001 | >3600 | 8 |
| soybean | 0.01 | 2604 | 6 |
| spect | 0.001 | 1365 | <1 |
| spect | 0.01 | 2269 | <1 |
| spect | 0.1 | - | 2 |
| vote | 0.001 | >3600 | 13 |
| vote | 0.01 | 2114 | 8 |
| vote | 0.1 | - | 9 |
| yeast | 0.001 | - | 16 |
| yeast | 0.01 | - | 13 |
| yeast | 0.1 | - | 21 |
| **Geometric mean improvement** | | | 197.27 |

## B.3 Memory Performance

Using the same experimental setup described in Sec. B.2, we additionally evaluated the memory usage of both methods. Tables 7–9 show the detailed results (in gigabytes). Datasets for which both methods required less than 1 GB of memory are omitted from the tables, although their values are included in the geometric means of the average memory usage ratios $m_{\text{TreeFARMS}}/m_{\text{SORTD}}$ reported at the end of each table.

Table 7: Memory usage (GB) of methods to calculate Rashomon sets across datasets with $\lambda = 0.001$ and $n^T = 10^n$, $n \in \{1, .., 6\}$. Results are reported as mean $\pm$ standard error. A '–' indicates results that are omitted due to their high memory requirement.

| Dataset | $|D|$ | $|F|$ | $d = 4$ | | $d = 5$ | |
|---|---|---|---|---|---|---|
| | | | TreeFARMS | SORTD | TreeFARMS | SORTD |
| coffee | 3816 | 15 | <1 | <1 | 1±0.0 | <1 |
| fico | 10459 | 17 | 1±0.0 | <1 | 8±0.0 | <1 |
| tic-tac-toe | 958 | 18 | <1 | <1 | 2±0.0 | <1 |
| spect | 267 | 22 | <1 | <1 | 3±0.0 | <1 |
| bank | 4521 | 23 | 1±0.0 | <1 | 8±0.0 | <1 |
| messidor | 1151 | 24 | <1 | <1 | 1±0.0 | <1 |
| hepatitis | 137 | 34 | 1±0.0 | <1 | 3±0.0 | <1 |
| kr-vs-kp | 3196 | 38 | 4±0.0 | <1 | 24±0.0 | <1 |
| hypothyroid | 3247 | 39 | 4±0.0 | <1 | 24±0.0 | <1 |
| soybean | 630 | 42 | 2±0.0 | <1 | 13±0.2 | <1 |
| anneal | 812 | 44 | 1±0.0 | <1 | 9±0.0 | <1 |
| yeast | 1484 | 46 | 10±0.1 | <1 | 24±0.0 | <1 |
| lymph | 148 | 47 | 1±0.0 | <1 | 4±0.1 | <1 |
| vote | 435 | 48 | 4±0.0 | <1 | 17±0.2 | <1 |
| HTRU2 | 17898 | 57 | 24±0.0 | <1 | 39±5.6 | <1 |
| biodeg | 1055 | 81 | 24±0.0 | <1 | 71±7.8 | <1 |
| haberman | 306 | 92 | 1±0.0 | <1 | 4±0.0 | <1 |
| **Geometric mean improvement** | | | | 24.95 | | 44.99 |

The geometric mean results indicate that, for each depth and $\lambda$ configuration, SORTD reduces memory usage by more than an order of magnitude compared to TreeFARMS. Combined with the runtime results, these findings demonstrate that SORTD achieves superior overall efficiency in both runtime and memory performance.

Across all datasets, maximum depths, and $\lambda$ values, SORTD's memory usage consistently remains below 1 GB. In contrast, TreeFARMS shows substantially higher memory requirements, particularly at greater depths for $\lambda = 0.001$ and $\lambda = 0.01$, and in datasets with larger feature spaces.

Table 8: Memory usage (GB) of methods to calculate Rashomon sets across datasets with $\lambda = 0.01$ and $n^T = 10^n$, $n \in \{1, .., 6\}$. Results are reported as mean $\pm$ standard error. A '−' indicates results that are omitted due to their high memory requirement.

| Dataset | $|D|$ | $|F|$ | $d = 4$ | | $d = 5$ | |
| --- | --- | --- | --- | --- | --- | --- |
| | | | TreeFARMS | SORTD | TreeFARMS | SORTD |
| coffee | 3816 | 15 | <1 | <1 | 1±0.0 | <1 |
| fico | 10459 | 17 | 1±0.0 | <1 | 8±0.0 | <1 |
| tic-tac-toe | 958 | 18 | <1 | <1 | 2±0.0 | <1 |
| spect | 267 | 22 | <1 | <1 | 3±0.0 | <1 |
| bank | 4521 | 23 | 1±0.1 | <1 | 4±0.5 | <1 |
| messidor | 1151 | 24 | <1 | <1 | 1±0.0 | <1 |
| hepatitis | 137 | 34 | 1±0.0 | <1 | 8±0.1 | <1 |
| kr-vs-kp | 3196 | 38 | 4±0.1 | <1 | 24±0.0 | <1 |
| hypothyroid | 3247 | 39 | 2±0.4 | <1 | 4±1.2 | <1 |
| soybean | 630 | 42 | 2±0.0 | <1 | 13±0.3 | <1 |
| anneal | 812 | 44 | 1±0.0 | <1 | 8±0.1 | <1 |
| yeast | 1484 | 46 | 10±0.1 | <1 | 24±0.0 | <1 |
| lymph | 148 | 47 | 2±0.0 | <1 | 12±0.2 | <1 |
| vote | 435 | 48 | 3±0.2 | <1 | 7±0.9 | <1 |
| HTRU2 | 17898 | 57 | 17±1.9 | <1 | 22±3.0 | 1±0.4 |
| biodeg | 1055 | 81 | 36±4.2 | <1 | 100±0.0 | <1 |
| haberman | 306 | 92 | 2±0.2 | <1 | 5±0.1 | <1 |
| **Geometric mean improvement** | | | | 21.28 | | 30.36 |

Table 9: Memory usage (GB) of methods to calculate Rashomon sets across datasets with $\lambda = 0.1$ and $n^T = 10^n$, $n \in \{1, .., 6\}$. Results are reported as mean $\pm$ standard error. A '−' indicates results that are omitted due to their high memory requirement.

| Dataset | $|D|$ | $|F|$ | $d = 4$ | | $d = 5$ | |
| --- | --- | --- | --- | --- | --- | --- |
| | | | TreeFARMS | SORTD | TreeFARMS | SORTD |
| kr-vs-kp | 3196 | 38 | 3±0.4 | <1 | 7±1.6 | <1 |
| yeast | 1484 | 46 | 3±0.7 | <1 | 6±1.6 | <1 |
| HTRU2 | 17898 | 57 | 2±0.7 | <1 | 2±0.7 | 5±1.7 |
| biodeg | 1055 | 81 | 9±1.8 | <1 | - | - |
| haberman | 306 | 92 | 1±0.1 | <1 | 2±0.4 | 1±0.2 |
| **Geometric mean improvement** | | | | 12.91 | | 4.21 |

## B.4 Rashomon multipliers

Table 10 reports the minimum Rashomon multiplier $\varepsilon$ values computed by SORTD to yield at least $10^n$ trees for each $n \in \{1, \ldots, 6\}$ for a subset of datasets. These results correspond to instances with $\lambda = 0.01$ and a depth budget of four. For readability, values are rounded to two decimal places.

The table reveals substantial variation in the Rashomon multipliers required to achieve a given set size across datasets. For example, while $\varepsilon = 0.16$ is necessary to obtain 10 trees for the *hypothyroid*

dataset, the same value yields over one million trees for the *spect* dataset. This illustrates the difficulty of selecting an appropriate Rashomon multiplier to target a desired set size. SORTD addresses this challenge through its ordered solution enumeration and anytime behavior: enumeration can be stopped once the desired number of models is reached, while still preserving the Rashomon set property—without requiring prior knowledge of the Rashomon multiplier (or bound).

Table 10: Rashomon multipliers required to obtain at least $10^n$ trees for instances with $\lambda = 0.01$ and depth budget four. The required multipliers vary strongly across datasets.

| Dataset | $n^T$ | | | | | |
|---|---|---|---|---|---|---|
| | $10^1$ | $10^2$ | $10^3$ | $10^4$ | $10^5$ | $10^6$ |
| bank | 0.08 | 0.13 | 0.16 | 0.23 | 0.25 | 0.32 |
| car | 0.03 | 0.06 | 0.11 | 0.14 | 0.17 | 0.23 |
| hypothyroid | 0.16 | 0.19 | 0.34 | 0.39 | 0.54 | 0.58 |
| monk1 | 0.0 | 0.14 | 0.29 | 0.43 | 0.43 | 0.57 |
| monk2 | 0.01 | 0.03 | 0.04 | 0.07 | 0.09 | 0.12 |
| monk3 | 0.06 | 0.14 | 0.21 | 0.25 | 0.35 | 0.42 |
| mouse | 0.0 | 0.0 | 0.08 | 0.19 | 0.27 | 0.33 |
| spect | 0.0 | 0.02 | 0.05 | 0.07 | 0.1 | 0.13 |
| tic-tac-toe | 0.01 | 0.01 | 0.04 | 0.05 | 0.08 | 0.1 |
| vote | 0.16 | 0.17 | 0.32 | 0.32 | 0.44 | 0.47 |

## B.5 Comparison of the tree ordering

A key difference between the state-of-the-art approach (TreeFARMS [30]) and SORTD lies in how trees are ordered within the Rashomon set. Fig. 10 illustrates this distinction. The figure reports the minimum number of trees that must be examined sequentially from the start of the Rashomon set to include all trees within the top $x\%$ of the lowest distinct objective values.

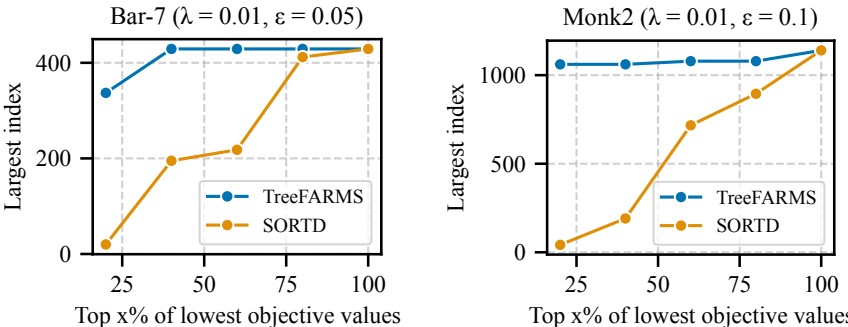

Figure 10: Largest index of the trees that belong to the top-$x\%$ of the lowest distinct objective values.

As the examples show, identifying trees within the top $20\%$ of lowest objective values using the state-of-the-art method may require exploring nearly the entire set. In contrast, SORTD provides early access to high-quality solutions, greatly reducing the overall evaluation effort.

## B.6 Runtime performance without trivial extensions

In Sec. B.2, we evaluated the runtime performance of SORTD while allowing trees with trivial extensions during Rashomon set computation. A trivial extension refers to a split that produces two leaf nodes with the same label (see Sec. A.8). We now assess how excluding such trees affects SORTD's runtime performance.

Following the same experimental setup as in Sec. B.2, we generated benchmark instances by varying the complexity penalty $\lambda \in \{0.001, 0.01, 0.1\}$ and the depth budget $d \in \{4, 5\}$. For each (dataset, $\lambda$, $d$) combination, SORTD was used to compute the minimum Rashomon multiplier (and

corresponding Rashomon bound) required to obtain at least $10^n$ trees for every $n \in \{1, \ldots, 6\}$. Using these multipliers, we constructed the respective Rashomon sets to evaluate the performance of SORTD against TreeFARMS.

Runtime results are presented in Tables 11–13. For a fixed Rashomon set size, ignoring trivial extensions can increase the required multiplier values, making some instances slightly more difficult. This is reflected in a modest rise in runtime. Nevertheless, SORTD's runtime performance remains stable, consistently achieving up to two orders of magnitude speed improvement over TreeFARMS.

Table 11: Runtime (s) of both methods for Rashomon set computation with trivial extensions ignored, across datasets and depth budgets, for $\lambda = 0.001$ and $n^T = 10^n$, $n \in \{1, \ldots, 6\}$. Results are reported as mean ± standard error.

| | | | $d = 4$ | | $d = 5$ | |
|---|---|---|---|---|---|---|
| Dataset | $|D|$ | $|F|$ | TreeFARMS | SORTD | TreeFARMS | SORTD |
| breast | 699 | 10 | <1 | <1 | <1 | <1 |
| monk2 | 169 | 11 | <1 | <1 | 3±0.1 | <1 |
| diabetes | 768 | 11 | <1 | <1 | <1 | <1 |
| compas | 6907 | 12 | <1 | <1 | 1±0.0 | <1 |
| bar-7 | 1913 | 14 | <1 | <1 | 1±0.1 | <1 |
| car | 1728 | 15 | 1±0.0 | <1 | 7±0.1 | <1 |
| expensive | 1417 | 15 | 1±0.0 | <1 | 6±0.1 | <1 |
| cheap | 2653 | 15 | 1±0.0 | <1 | 5±0.1 | <1 |
| monk1 | 124 | 15 | <1 | <1 | 6±0.8 | <1 |
| monk3 | 122 | 15 | 3±0.2 | <1 | 31±3.7 | <1 |
| coffee | 3816 | 15 | 1±0.0 | <1 | 6±0.2 | <1 |
| banknote | 1372 | 16 | <1 | <1 | <1 | <1 |
| tumor | 336 | 17 | 2±0.0 | <1 | 18±0.1 | <1 |
| fico | 10459 | 17 | 6±0.1 | <1 | 38±0.8 | 2±0.1 |
| tic-tac-toe | 958 | 18 | 2±0.0 | <1 | 22±0.3 | <1 |
| spect | 267 | 22 | 29±1.2 | <1 | >300 | <1 |
| bank | 4521 | 23 | 11±0.2 | <1 | 121±1.7 | 1±0.0 |
| messidor | 1151 | 24 | 5±0.2 | <1 | 46±1.6 | <1 |
| hepatitis | 137 | 34 | 112±3.6 | <1 | >300 | 2±0.1 |
| kr-vs-kp | 3196 | 38 | 62±2.7 | <1 | >300 | 10±0.3 |
| hypothyroid | 3247 | 39 | 58±1.8 | <1 | >300 | 8±0.4 |
| soybean | 630 | 42 | 65±1.7 | <1 | >300 | 6±0.2 |
| anneal | 812 | 44 | 42±1.6 | <1 | >300 | 5±0.2 |
| mouse | 70 | 45 | 12±0.6 | <1 | >300 | 4±0.2 |
| yeast | 1484 | 46 | 240±4.2 | <1 | >300 | 16±0.4 |
| lymph | 148 | 47 | 163±8.1 | <1 | >300 | 8±0.5 |
| vote | 435 | 48 | >300 | <1 | >300 | 9±0.1 |
| HTRU2 | 17898 | 57 | >300 | 4±0.1 | >300 | 111±2.9 |
| biodeg | 1055 | 81 | >300 | 4±0.1 | >300 | 194±6.9 |
| haberman | 306 | 92 | >300 | 2±0.1 | >300 | 147±4.3 |
| **Geometric mean improvement** | | | | 95.29 | | 32.90 |

Table 12: Runtime (s) of both methods for Rashomon set computation with trivial extensions ignored, across datasets and depth budgets, for $\lambda = 0.01$ and $n^T = 10^n$, $n \in \{1, \ldots, 6\}$. Results are reported as mean ± standard error.

| | | | $d = 4$ | | $d = 5$ | |
|---|---|---|---|---|---|---|
| Dataset | $|D|$ | $|F|$ | TreeFARMS | SORTD | TreeFARMS | SORTD |
| breast | 699 | 10 | <1 | <1 | <1 | <1 |
| monk2 | 169 | 11 | <1 | <1 | 3±0.1 | <1 |
| diabetes | 768 | 11 | <1 | <1 | <1 | <1 |

| Dataset | $|D|$ | $|F|$ | $d = 4$ TreeFARMS | $d = 4$ SORTD | $d = 5$ TreeFARMS | $d = 5$ SORTD |
|---|---|---|---|---|---|---|
| compas | 6907 | 12 | <1 | <1 | 1±0.2 | <1 |
| bar-7 | 1913 | 14 | <1 | <1 | 1±0.1 | <1 |
| car | 1728 | 15 | 1±0.1 | <1 | 7±0.2 | <1 |
| expensive | 1417 | 15 | 1±0.0 | <1 | 7±0.1 | <1 |
| cheap | 2653 | 15 | 1±0.1 | <1 | 4±0.4 | <1 |
| monk1 | 124 | 15 | 2±0.2 | <1 | 14±2.6 | <1 |
| monk3 | 122 | 15 | 3±0.3 | <1 | 54±5.1 | <1 |
| coffee | 3816 | 15 | 1±0.0 | <1 | 6±0.2 | <1 |
| banknote | 1372 | 16 | <1 | <1 | <1 | <1 |
| tumor | 336 | 17 | 2±0.1 | <1 | 18±0.3 | <1 |
| fico | 10459 | 17 | 6±0.3 | <1 | 43±1.3 | 2±0.2 |
| tic-tac-toe | 958 | 18 | 2±0.0 | <1 | 22±0.4 | <1 |
| spect | 267 | 22 | 38±1.5 | <1 | >300 | <1 |
| bank | 4521 | 23 | 10±0.6 | <1 | 75±7.6 | 2±0.2 |
| messidor | 1151 | 24 | 6±0.3 | <1 | 38±2.1 | <1 |
| hepatitis | 137 | 34 | 134±3.0 | <1 | >300 | 1±0.1 |
| kr-vs-kp | 3196 | 38 | 70±3.1 | <1 | >300 | 10±0.4 |
| hypothyroid | 3247 | 39 | 37±5.3 | <1 | 138±26.1 | 6±0.6 |
| soybean | 630 | 42 | 80±2.2 | <1 | >300 | 5±0.2 |
| anneal | 812 | 44 | 47±2.6 | <1 | >300 | 5±0.2 |
| mouse | 70 | 45 | 15±0.8 | <1 | >300 | 5±0.2 |
| yeast | 1484 | 46 | 234±5.6 | 1±0.1 | >300 | 13±0.5 |
| lymph | 148 | 47 | >300 | <1 | >300 | 6±0.2 |
| vote | 435 | 48 | >300 | <1 | >300 | 7±0.5 |
| HTRU2 | 17898 | 57 | 223±25.6 | 6±0.4 | 227±25.2 | 126±6.7 |
| biodeg | 1055 | 81 | >300 | 4±0.2 | >300 | 98±0.9 |
| haberman | 306 | 92 | >300 | 3±0.2 | >300 | 122±2.1 |
| **Geometric mean improvement** | | | | 70.30 | | 29.77 |

Table 13: Runtime (s) of both methods for Rashomon set computation with trivial extensions ignored, across datasets and depth budgets, for $\lambda = 0.1$ and $n^T = 10^n$, $n \in \{1, \ldots, 6\}$. Results are reported as mean ± standard error. A "–" indicates runs omitted due to excessive memory usage.

| Dataset | $|D|$ | $|F|$ | $d = 4$ TreeFARMS | $d = 4$ SORTD | $d = 5$ TreeFARMS | $d = 5$ SORTD |
|---|---|---|---|---|---|---|
| breast | 699 | 10 | <1 | <1 | <1 | <1 |
| monk2 | 169 | 11 | <1 | <1 | 2±0.4 | <1 |
| diabetes | 768 | 11 | <1 | <1 | <1 | <1 |
| compas | 6907 | 12 | <1 | <1 | 1±0.2 | <1 |
| bar-7 | 1913 | 14 | <1 | <1 | 1±0.3 | <1 |
| car | 1728 | 15 | 1±0.2 | <1 | 4±0.9 | <1 |
| expensive | 1417 | 15 | <1 | <1 | 2±0.7 | <1 |
| cheap | 2653 | 15 | <1 | <1 | 2±0.5 | <1 |
| monk1 | 124 | 15 | 2±0.3 | <1 | 17±3.9 | <1 |
| monk3 | 122 | 15 | 2±0.5 | <1 | 24±6.7 | <1 |
| coffee | 3816 | 15 | 1±0.2 | <1 | 3±0.7 | <1 |
| banknote | 1372 | 16 | <1 | <1 | <1 | <1 |
| tumor | 336 | 17 | 1±0.2 | <1 | 4±1.4 | <1 |
| fico | 10459 | 17 | 4±0.8 | <1 | 9±2.6 | 2±0.3 |
| tic-tac-toe | 958 | 18 | 2±0.3 | <1 | 6±1.6 | <1 |
| spect | 267 | 22 | 17±3.1 | <1 | 90±25.5 | <1 |
| bank | 4521 | 23 | 4±1.0 | <1 | 8±2.4 | 2±0.3 |
| messidor | 1151 | 24 | 2±0.7 | <1 | 4±1.2 | 1±0.2 |

|          |       |       | $d = 4$         |        | $d = 5$         |         |
|----------|-------|-------|-----------------|--------|-----------------|---------|
| Dataset  | $|D|$ | $|F|$ | TreeFARMS       | SORTD  | TreeFARMS       | SORTD   |
| hepatitis   | 137   | 34 | 34±10.0  | <1    | 41±13.0  | 2±0.3  |
| kr-vs-kp    | 3196  | 38 | 53±7.9   | <1    | 178±28.2 | 5±0.8  |
| hypothyroid | 3247  | 39 | 21±6.8   | <1    | 38±13.7  | 6±0.9  |
| soybean     | 630   | 42 | 31±8.9   | <1    | 42±12.1  | 5±0.8  |
| anneal      | 812   | 44 | 11±3.4   | <1    | 16±5.3   | 3±0.5  |
| mouse       | 70    | 45 | 21±6.3   | <1    | 110±28.2 | 1±0.3  |
| yeast       | 1484  | 46 | 114±16.0 | 1±0.2 | 185±24.7 | 10±1.5 |
| lymph       | 148   | 47 | 112±25.0 | <1    | 122±26.8 | 4±0.7  |
| vote        | 435   | 48 | 64±24.1  | <1    | 67±24.7  | 5±1.1  |
| HTRU2       | 17898 | 57 | 49±13.5  | 5±0.7 | 70±22.0  | 50±9.2 |
| biodeg      | 1055  | 81 | 177±26.9 | 6±0.8 | -        | -      |
| haberman    | 306   | 92 | 259±21.4 | 2±0.3 | 276±23.1 | 22±3.7 |
| **Geometric mean improvement** | | | | 28.30 | | 11.71 |

## B.7 Variable importance analysis

Variable importance on Rashomon sets was evaluated using the *leave one feature out* procedure [50] with $\lambda = 0.01$, depth $d = 4$. For each dataset, we generated 20 stratified bootstrap samples, computed the Rashomon multiplier required to obtain at least 10,000 trees, and constructed the corresponding set using these multipliers. For each bootstrap sample, we also calculated the top-1 and top-100 Rashomon sets and ranked features according to the increase in area under the tree-index versus objective-value curve when each feature was removed. Kendall's $\tau$ and Jaccard index were used to evaluate the stability of the full and top-5 variable ranking, respectively. Table 14 presents the results.

Table 14: Variable importance similarity compared to LOFO based on $n^T = 10^4$.

|          | Kendall's $\tau$ | | Jaccard Index | |
|----------|-----------|-------------|-----------|-------------|
| Dataset  | $n^T = 1$ | $n^T = 100$ | $n^T = 1$ | $n^T = 100$ |
| diabetes    | 0.71  | 0.89 | 1.0 | 1.0 |
| anneal      | 0.02  | 0.59 | 0.8 | 1.0 |
| bank        | -0.22 | 0.64 | 0.4 | 1.0 |
| banknote    | 0.68  | 0.93 | 1.0 | 1.0 |
| bar-7       | 0.56  | 0.78 | 1.0 | 1.0 |
| breast      | 0.69  | 0.91 | 0.6 | 1.0 |
| car         | 0.70  | 0.83 | 0.8 | 0.8 |
| cheap       | 0.47  | 0.83 | 0.8 | 1.0 |
| coffee      | 0.73  | 0.81 | 1.0 | 1.0 |
| compas      | 0.42  | 0.88 | 0.6 | 0.8 |
| expensive   | 0.79  | 0.79 | 0.8 | 0.6 |
| fico        | 0.13  | 0.79 | 0.4 | 0.8 |
| haberman    | 0.25  | 0.46 | 0.4 | 0.8 |
| hepatitis   | 0.43  | 0.62 | 0.2 | 0.4 |
| HTRU2       | 0.12  | 0.21 | 0.6 | 1.0 |
| hypothyroid | 0.11  | 0.42 | 0.4 | 0.8 |
| kr-vs-kp    | 0.39  | 0.49 | 0.8 | 0.8 |
| lymph       | 0.50  | 0.72 | 0.8 | 0.8 |
| messidor    | 0.60  | 0.80 | 0.8 | 1.0 |
| monk1       | 0.66  | 0.66 | 0.6 | 0.8 |
| monk2       | 0.75  | 0.75 | 1.0 | 0.8 |
| monk3       | 0.68  | 0.64 | 1.0 | 1.0 |
| mouse       | 0.34  | 0.51 | 1.0 | 0.8 |
| tumor       | 0.57  | 0.85 | 0.8 | 1.0 |
| soybean     | 0.26  | 0.79 | 0.8 | 1.0 |

| Dataset | Kendall's $\tau$ | | Jaccard Index | |
|---|---|---|---|---|
| | $n^T = 1$ | $n^T = 100$ | $n^T = 1$ | $n^T = 100$ |
| spect | 0.58 | 0.74 | 0.8 | 0.6 |
| tic-tac-toe | 0.61 | 0.67 | 1.0 | 1.0 |
| vote | 0.06 | 0.69 | 0.2 | 0.8 |
| yeast | 0.38 | 0.74 | 0.8 | 0.8 |

The stability of the top-1 tree versus the top-10,000 trees is low for most datasets with only five of them satisfying $\tau \geq 0.7$, and 19 of them have a Jaccard index of value at least $0.8$. In contrast, for the top-100 versus top-10,000 trees, 17 datasets satisfy $\tau \geq 0.7$ and 26 exceed a Jaccard index exceeding $0.8$, indicating that smaller Rashomon sets can yield similar variable importance values as larger Rashomon sets.

## B.8 Regression results

This section provides the details on the regression experiment shown in Fig. 7 and additionally analyses the runtime performance of SORTD for regression.

**Data** For the regression experiments, we use the datasets from Zhang et al. [47] and also follow their binarization of non-binary features.[3] The datasets can also be obtained from the UCI Machine Learning repository [55]. For all datasets, we normalize the regression label by subtracting the mean and dividing by the standard deviation.

**Comparison with CART and STreeD** Since there are, to the best of our knowledge, no previous methods to enumerate the whole Rashomon set of regression trees, we follow the set-up by Xin et al. [30], who did a similar experiment for classification trees: we compare SORTD with both the heuristic CART [2] and the optimal method STreeD [40] when those are called repeatedly on random samples of the data. STreeD is a state-of-the-art optimal method for computing optimal regression trees [58].[4]

In this experiment, we impose a 60 seconds timeout for each method. CART and STreeD are run repeatedly within that time budget on random samples of 50% of the total dataset. All resulting unique trees are kept. SORTD, on the other hand, is run once and collects trees in the Rashomon set in order until time-out or until the Rashomon set is exhausted. We use the Rashomon multiplier $\varepsilon = 0.1$, complexity cost $\lambda = 0.001$, and maximum depth $d = 4$.

Fig. 7 in the main text shows how SORTD can find orders of magnitude more trees in the Rashomon set than either CART or STreeD within the time limit. For the *Synchronous Machine Dataset*, SORTD finds the whole Rashomon set within two seconds, whereas both STreeD and CART find only a fraction in the allotted 60 seconds. For both *Airfoil Self-Noise* and *Seoul Bike Sharing Demand*, SORTD finds orders of magnitude more trees than either CART or STreeD in the given time limit. These results confirm what Xin et al. [30] also concluded for classification: enumerating the whole Rashomon set can be done best with a dedicated method.

**Runtime** Additionally, we analyse the runtime performance of SORTD to calculate the Rashomon set of regression trees. Here, we set the complexity cost $\lambda = 0.01$, the Rashomon multiplier to $\varepsilon = 1.0$, the maximum number of trees $n^T = 10^6$, and we test both with a maximum depth of four and five. We run all experiments five times on all datasets with a time-out of 300 seconds.

Table 15 shows that SORTD successfully enumerates the top one million trees for all benchmark datasets within the time limit for $d = 4$, except for one, and for all but three for $d = 5$.

In comparison to the experiments on classification trees shown above in Appendix B.2, the regression tree Rashomon set is more time intensive to compute. We think this is because the regression loss allows for many more unique loss values. Since SORTD combines solutions with the same value (see Appendix A.2), more unique loss values result in a higher runtime. Runtime performance can

---

[3] https://github.com/ruizhang1996/regression-tree-benchmark
[4] https://github.com/algtudelft/pystreed

Table 15: SORTD runtime (s) performance of methods to calculate Rashomon sets for regression trees across datasets with $\lambda = 0.01$ and $n^T = 10^6$. Results are reported as mean $\pm$ standard error.

| Dataset | $|D|$ | $|F|$ | $d = 4$ | $d = 5$ |
|---|---|---|---|---|
| Airfoil Self-Noise | 1503 | 17 | $2 \pm 0.1$ | $2 \pm 0.1$ |
| Air Quality | 111 | 16 | $2 \pm 0.3$ | $9 \pm 0.9$ |
| Energy Efficiency (Cooling) | 768 | 27 | $80 \pm 5.0$ | $> 300$ |
| Energy Efficiency (Heating) | 768 | 27 | $94 \pm 3.4$ | $> 300$ |
| Household | 2049280 | 15 | $53 \pm 0.1$ | $134 \pm 1.0$ |
| Medical Cost Personal | 1338 | 16 | $2 \pm 0.0$ | $12 \pm 0.1$ |
| Optical Interconnection Network | 640 | 29 | $< 1$ | $13 \pm 0.0$ |
| Real Estate Valuation | 414 | 18 | $13 \pm 0.0$ | $18 \pm 0.1$ |
| Seoul Bike Sharing Demand | 8760 | 32 | $> 300$ | $> 300$ |
| Servo | 167 | 15 | $2 \pm 0.1$ | $4 \pm 0.2$ |
| Synchronous Machine | 557 | 12 | $< 1$ | $< 1$ |
| Yacht Hydrodynamics | 308 | 35 | $14 \pm 0.5$ | $157 \pm 4.9$ |

likely be improved by setting a higher tolerance for which solution values are considered the 'same'. Currently, SORTD uses a tolerance of $10^{-4}$.

## B.9 Equality-of-opportunity results

This section provides the details on the equality-of-opportunity experiment shown in Fig. 8 and additionally analyses the runtime performance of SORTD for evaluating such a second objective.

**Data**  For the equality-of-opportunity experiments, we use the datasets from Le Quy et al. [48] and follow their suggested binarization of non-binary features. In Table 16, we report which feature is selected as the discrimination sensitive feature. References to the original datasets can be found here [51, 55, 59–63].

**Pareto front**  Fig. 8 shows how SORTD can be used to generate a Rashomon set for sparse classification trees, which are then evaluated using a secondary objective: equality of opportunity. Since SORTD generates the solutions in increasing order of its objective, it examines the most accurate trees first. Therefore, when halting the search at any time, a (partial) Pareto front over the primary and secondary objectives can be computed. The results in Fig. 8 show that among the top $10^7$ sparse depth-four classification trees for the *Adult* dataset, all trees have at least 2% discrimination, i.e., the true positive rate of one group is at least 2% higher than another. For the *Bank Marketing* dataset, a tree with zero discrimination is found among the top $10^7$ trees. For the *Compas* dataset the most fair dataset in the top $10^7$ trees still has 2.5% discrimination. It also shows that by reducing accuracy by only 0.5%, the discrimination score can be lowered from 7% to 2.5%.

**Runtime comparison**  We evaluate SORTD's efficiency in evaluating a second objective such as equality of opportunity by comparing it with the optimal dynamic programming approach STreeD [40], which supports optimizing accuracy and equality of opportunity. We do not compare with the mixed-integer programming approach by Jo et al. [64] since Linden et al. [40] report runtimes several orders of magnitude lower than theirs, while computing the same optimal solutions.

We run SORTD by iteratively generating the next best $n^T$ trees according to the regularized accuracy objective. We then evaluate the newly generated trees as described in Appendix A.9 using the equality-of-opportunity metric. If a tree is found within the pre-set discrimination limit $\delta$, we stop. Otherwise, SORTD generates the next batch of $n^T$ trees and repeats. Since SORTD yields trees in order of the regularized accuracy objective, the first tree it finds within the discrimination limit must be optimal with respect to the regularized accuracy objective.

In the comparison with STreeD, there are a couple of small differences that may influence runtime: (1) We run SORTD using the regularized accuracy objective, so each leaf node is penalized with

the sparsity penalty $\lambda$. In this experiment, we set $\lambda = 0.01$. STreeD's equality-of-opportunity optimization task by default does not consider such regularization. (2) STreeD also considers non-majority labels in leaf nodes, whereas SORTD by default does not. Because of these two differences, STreeD and SORTD are not guaranteed to find the same optimal solution, although we verified that the solutions are close. Since our main aim in this experiment is to validate that SORTD can successfully be used to post-evaluate a secondary objective, we consider these differences acceptable.

Table 16: Runtime (s) performance to compute an optimal fair tree with at most $1\%$ discrimination, and maximum depth $d = 3$, averaged over five runs. For SORTD, we set $\lambda = 0.01$ and iteratively evaluate $10^5$ trees. Results are reported as mean $\pm$ standard error. Best results are bold.

| Dataset | Sensitive feature | $|D|$ | $|F|$ | STreeD | SORTD |
|---|---|---|---|---|---|
| Adult | Gender | 45222 | 18 | $\mathbf{1 \pm 0.0}$ | $\mathbf{1 \pm 0.0}$ |
| Bank marketing | Married | 45211 | 47 | $8 \pm 0.1$ | $\mathbf{7 \pm 0.0}$ |
| Communities & crime | Race | 1994 | 98 | $\mathbf{6 \pm 0.0}$ | $21 \pm 0.1$ |
| Compas recid. | Race | 6172 | 10 | $\mathbf{<1}$ | $\mathbf{<1}$ |
| Compas viol. recid. | Race | 4020 | 10 | $\mathbf{<1}$ | $1 \pm 0.0$ |
| Diabetes | Race | 45715 | 129 | $\mathbf{36 \pm 0.0}$ | $58 \pm 0.2$ |
| Dutch census | Gender | 60420 | 59 | $\mathbf{9 \pm 0.0}$ | $79 \pm 0.5$ |
| German credit | Gender | 1000 | 70 | $32 \pm 0.2$ | $\mathbf{3 \pm 0.0}$ |
| KDD census income | Race | 284556 | 118 | $\mathbf{27 \pm 0.1}$ | $134 \pm 0.6$ |
| Lawschool | Race | 20798 | 20 | $\mathbf{<1}$ | $4 \pm 0.0$ |
| OULAD | Gender | 21562 | 46 | $17 \pm 0.2$ | $\mathbf{7 \pm 0.0}$ |
| Ricci | Race | 118 | 5 | $\mathbf{<1}$ | $\mathbf{<1}$ |
| Student Portuguese | Gender | 649 | 56 | $\mathbf{1 \pm 0.0}$ | $2 \pm 0.0$ |
| Student mathematics | Gender | 395 | 56 | $\mathbf{<1}$ | $6 \pm 0.0$ |

Table 16 shows the mean runtime performance of STreeD and SORTD to find fair and optimal trees with max-depth $d = 3$, discrimination limit $\delta = 1\%$, and sparsity penalty $\lambda = 0.01$ (for SORTD). With SORTD, we iteratively produce and evaluate $n^T = 10^5$ trees until a tree within the discrimination limit is found, or until the time-out of 300 seconds. The results show that SORTD remains close in performance to STreeD, and for some datasets, such as *German credit* and *OULAD*, even performs significantly better. These results are obtained by evaluating the secondary objective using callbacks to Python. Hence, if runtime performance was the main concern, these results could be further improved by computing this also in C++.

Concluding, these results show that using Rashomon sets to optimize one (totally ordered) objective, and later evaluating the Rashomon set (possibly iteratively) using a second objective (such that this multi-objective optimization task is only partially ordered) is promising.

