# OpenReview forum: "SORTeD Rashomon Sets of Sparse Decision Trees: Anytime Enumeration"
_NeurIPS.cc/2025/Conference — NeurIPS 2025 spotlight_

### Official Review · Reviewer_hQ9P · 2025-06-19

**Clarity:** 4
**Significance:** 3
**Originality:** 3
**Rating:** 5
**Confidence:** 4

**Summary:**

This work gives a method to enumerate the Rashomon set of sparse decision trees, over any totally ordered objective, in order of the trees' dataset objective value. The algorithm stores partial solutions at descending levels of the tree in two types of 'helper' nodes -- branching nodes, which enable further splits, and leaf nodes, which terminate the tree. Broadly speaking, the algorithm computes optimal solutions to subproblems in every branching node, and prunes subproblems by keeping running upper and lower bounds on the possible objective in a subproblem. The enumeration algorithm outputs the optimal tree at the top level (the global optimal tree), then in some sense "percolates" optimal solutions from the left and right subtrees up to the top of the tree. The algorithm stops when it has either enumerated all trees in the Rashomon set, has reached a time limit, or when it has enumerated a fixed number of trees. Thus, this algorithm gives two novel capabilities; first, it can enumerate the Rashomon set of sparse decision trees in order; second, it can compute the Rashomon set with respect to any totally ordered (and, for practical purposes, separable) objective function. The authors also include timing experiments to demonstrate an improvement over the current state-of-the-art in TreeFARMS, as well as case studies in variable importance, Rashomon sets of sparse regression trees, and fairness.

**Questions:**

1. Do you think that SORTD's speed advantage would persist if compared against TreeFARMS, but with the depth-2 solver? To rephrase -- is it the optimization framework itself that gives the advantage over TreeFARMS, or the incorporation of a very helpful extra algorithmic component?

2. Is this algorithm able to easily exclude trivial extensions from the Rashomon set (as defined in the TreeFARMS algorithm; extra splits that don't change predictions, but also don't exceed the Rashomon bound)? Ignoring trivial extensions is a useful feature for TreeFARMS which also aids computation, so comparing to TreeFARMS without ignoring trivial extensions appears to be hamstringing TreeFARMS unduly, if SORTD is in fact unable to easily ignore them.

3. How does the Rashomon set of sparse regression trees compare to the Rashomon set of sparse decision trees (say, on a threshold of the continuous regression label to turn it into a classification problem)? Are the Rashomon sets similar? Do the trees share many common prefixes?

**Ethical Concerns:**

["NO or VERY MINOR ethics concerns only"]

**Final Justification:**

After my discussion with the authors, I am raising my score to a 5 (accept). The actual method of the paper, anytime enumeration of the Rashomon set with respect to any separable objective function, is a significant step forward in the field of Rashomon set computation. My primary concerns with the paper were that the case studies presented as evaluation were superfluous and did not demonstrate new capabilities beyond what previously existed. After consideration of the responses and other reviews, I've concluded that the case studies sufficiently demonstrate that the proposed method can indeed handle flexible objective functions, is faster than TreeFARMS, and allows for computation with a fixed number of trees in addition to with a fixed Rashomon bound. In essence -- SORTD hasn't lost anything that TreeFARMS has, and it's faster and more flexible. With that in mind, the case studies are good enough for me to confidently recommend acceptance of the paper, given its strong methodological progress.

**Limitations:**

yes

**Quality:**

3

**Strengths And Weaknesses:**

## Strengths

1. The insight that the STreeD framework for decision tree optimization can be adapted to find all models in the Rashomon set of totally ordered (and perhaps additive) objective functions in order, and quickly at that, strikes me as both novel and significant.

2. The speedups over TreeFARMS, in combination with the anytime enumeration capability and objective function flexibility, are fairly represented by this paper to be steps toward the practical application of Rashomon set methods.

3. The paper was succinct, well-written, and easy to follow. The paper was well-situated among other literature in the related work section. The descriptions of the algorithms, both in the high-level idea and in the details in the paper and the appendix, were clear and I feel as if I understand both their motivation and implementation.


## Weaknesses

### Significant Weaknesses

In summary, I felt as if the case studies provided to demonstrate the practical utility of SORTD were somewhat superficial.

1. The Model Class Reliance metric due to Fisher et al., 2019, already captures the essence of the LOFO case study. The resulting insights in the first example that "priors > 3" and "ExternalRiskEstimate" are important are not new, and are easily obtainable via existing approaches.

2. The work of Donnelly et al., 2023: The Rashomon Importance Distribution: Getting RID of Unstable, Single Model-based Variable Importance gives a principled way to obtain a distribution of variable importances in the Rashomon set via bootstrapping.
In the second example, I feel as if the novel insight from this paper is not that Rashomon sets improve the stability of variable importance (which is thoroughly discussed by Donnelly et al., 2023), but that you don't need many models from the Rashomon set to significantly improve reliability. This point felt under-explored -- indeed, only 20/29 of the datasets analyzed met the 0.8 threshold for Jaccard index for top-100, which to me is not enough to indicate reliability.

3. Similarly, the example given for generating sparse regression trees, while interesting for the fact that SORTD can indeed compute the Rashomon set of sparse regression trees, did not add much to the paper. TreeFARMS was already shown by Xin et al., 2022 to dominate methods which generate single optimal trees on bootstraps, and there is no reason to expect that another Rashomon set algorithm should be different. I think that the main point of this section was that SORTD can indeed effectively compute the Rashomon set of objective functions beyond misclassification error, and perhaps this could have been emphasized moreso than comparison to CART and STreeD.

### Minor Weaknesses

1. The abstract seems to imply that SORTD somehow bypasses the NP-hard optimal tree problem by enumerating Rashomon sets in order. This is an issue of wording, not content.

2. I think that this work would benefit from a figure or analysis supporting the claim on line 243, that the scalability of SORTD derives from the shortening of solution lists, despite more branching nodes. This analysis seems plausible when adding bad or do-nothing features to the dataset, but when adding good features I would expect the number of models in the Rashomon set to explode, and concurrently for solution lists to get longer even though there are more branching nodes.

---

> ### Author Rebuttal · Authors · 2025-07-31
>
> We would like to thank the reviewer for the positive comments. We reply to the questions below:
>
> **Q1:** *"Do you think that SORTD's speed advantage would persist if compared against TreeFARMS, but with the depth-2 solver?..."*
>
> The scalability advantage is indeed mostly obtained through the depth-two solver. The novel search technique provides an anytime behavior, which we believe is essential to parse larger datasets, specifically, if beforehand good Rashomon bounds cannot be guessed. Compared to TreeFARMS, SORTD is also more flexible in terms of the variety of objective functions it supports.
>
> **Q2:** *"...comparing to TreeFARMS without ignoring trivial extensions appears to be hamstringing TreeFARMS unduly..."*
>
>  The impact of ignoring trivial extensions is a much smaller Rashomon set (under the assumption of a fixed Rashomon multiplier epsilon values), thus improving both methods' runtime. In our experiments, on the other hand, we search for fixed-size Rashomon sets (e.g., 10^6, see Table 2). After modifying SORTD to support the exclusion of trivial extensions, our evaluation shows that both methods had higher runtime compared to the case where trivial extensions are not ignoried (as fixing Rashomon set size resulted in higher Rashomon multiplier epsilon values). However, SORTD still maintained up to a two orders of magnitude runtime advantage over TreeFARMS. We will include the results in the appendix.
>
> **Q3:** *"How does the Rashomon set of sparse regression trees compare to the Rashomon set of sparse decision trees"*
>
> The main difference we observed was that the number of unique solution values for regression was much larger than for classification. This is expected since the regression loss is continuous. Investigating the comparison of these two sets could be interesting for future research.
>
> **SW1:** *"The Model Class Reliance metric due to Fisher et al., 2019, already captures the essence of the LOFO case study..."*
>
> Our objective in Figure 5 is to give a visualization of how we calculated LOFO values using Rashomon sets rather than providing an analysis of certain datasets. We will make it clear in the paper.
>
> **SW2:** *"I feel as if the novel insight from this paper is not that Rashomon sets improve the stability of variable importance ... but that you don't need many models from the Rashomon set to significantly improve reliability..."*
>
> Our variable importance analysis indeed does not aim to demonstrate that Rashomon sets improve the stability of variable importance. It aims to show that variable importance measured is relatively stable for 100 or 10,000 trees. Yet, developing theories on the necessary number of trees, rather than the bound (as in Donnelly et al., 2023), to get stable measurements of variable importance is an interesting future work.
>
> **SW3:** *"...I think that the main point of this section was that SORTD can indeed effectively compute the Rashomon set of objective functions beyond misclassification error..."*
>
> Indeed, in Section 5.3, we aim to demonstrate the capability of SORTD to calculate and evaluate Rashomon sets of varying objectives. As there is no benchmark method of generating Rashomon sets of regression trees, we demonstrate SORTD’s ability by comparing it with CART. Highlighting this in the text is useful; we will make the necessary adjustments.
>
> **MW1:** *"The abstract seems to imply that SORTD somehow bypasses the NP-hard optimal tree problem..."*
>
> Thank you, we will change the sentence.
>
> **MW2:** *"...when adding good features I would expect the number of models in the Rashomon set to explode, and concurrently for solution lists to get longer..."*
>
> We would like to highlight that the size of the Rashomon set is fixed in this analysis. Adding good features would potentially increase the Rashomon set size given a fixed Rashomon bound. However, given a fixed Rashomon set size, the solutions would be distributed to the solutions list of more branching nodes.

---

> > ### Comment · Reviewer_hQ9P · 2025-08-04
> >
> > Thank you for the thorough and thoughtful response.
> >
> > **Q1:** I am satisfied by this response regarding the speed-up; it is useful to know that the speed advantage is mostly creditable to the depth-2 solver. The flexibility of objective function seems immediately relevant and useful. I am unconvinced that the anytime behavior changes the story -- I find it roughly equivalently useful to say, "I want all models within 1% of the best objective" rather than "I want the top 1000 models."
> >
> > **Q2:** This response has proven to me that SORTD can in fact easily ignore trivial extensions, which solves my concern. I think the paper will benefit from these results in the appendix.
> >
> > **Q3:** Thank you, this is interesting.
> >
> > **SW1:** I apologize, I should have been more clear. My concern is that the metric proposed in the case study is not a novel contribution, because of the model class reliance metric established in prior literature. When criticizing the particulars of the dataset, my intention was to say that the kinds of results obtained via this method are not different than with prior methods. With these in mind, I still feel as if this case study does not add much to prior work.
> >
> > **SW2:** Thank you for the response, I understand the purpose of the case study better. With that in mind, I do not feel as if the results of the case study support the conclusion -- 20/29 datasets meeting the (arbitrarily chosen) threshold is not indicative to me of reliability. I think the proposed direction of future work could be very helpful towards strengthening the results in this section.
> >
> > **SW3:** I think highlighting this in the text is sufficient to provide clarity.
> >
> > I am satisfied by both answers to the minor weaknesses.

---

> > > ### Author Response · Authors · 2025-08-05
> > >
> > > Thank you for your suggestions to improve our work. Below are our clarifications regarding the points you raised:
> > >
> > > **Q1:** *I find it roughly equivalently useful to say, "I want all models within 1% of the best objective" rather than "I want the top 1000 models."*
> > >
> > > We agree that bounding the Rashomon set by a fixed (relative) bound and by the number of top models are both useful. However, they are not equivalent, as we show in Table 5 in the supplementary material. E.g., setting $\varepsilon = 0.16$ yields only 10 trees for the *hypothyroid* dataset but over one million trees for the *spect* dataset.
> > >
> > > Moreover, the anytime behaviour achieved by bounding the Rashomon set by the number of top models is particularly important to parse larger datasets. It allows us to obtain results within a few minutes without waiting for the complete enumeration of the Rashomon set under a given epsilon.
> > >
> > > Therefore, we believe that the ability to bound the Rashomon set by a fixed size further extends and enables the exploration and application of the Rashomon set paradigm.
> > >
> > >
> > >
> > > **SW1 & SW2:** *"I still feel as if this case study does not add much to prior work"*, *"I do not feel as if the results of the case study support the conclusion"*
> > >
> > >
> > > We agree that the conclusion (lines 267 to 268) could better reflect our intention of highlighting the potential of exploring variable importance across different Rashomon set sizes. We will clarify this point in the text, and highlight the potential for bounding the Rashomon set by its size (enabled by our approach) for further theoretical and empirical future work.

---

> > > > ### Comment · Reviewer_hQ9P · 2025-08-05
> > > >
> > > > Thank you for the clarifications; I am satisfied by the SW1 and SW2 proposal, and I've responded to the Q1 discussion below.
> > > >
> > > > **Q1:** I am not arguing that the two ideas are equivalent, rather equivalently useful. From a practitioner's standpoint, I find it appealing to establish an error tolerance which I am willing to accept, then simply find all trees within that tolerance. The top N trees may be more or less meaningful, depending on the particular situation or preference of the practitioner. One may obtain a similar speedup in computation by using a smaller epsilon. Top N enumeration, in my view, gives a new tool in the toolbox of a practitioner using the Rashomon set paradigm to approach a learning problem, but it does not enable applications or exploration which were not possible before. The flexibility of objective function and the speed-up, on the other hand, I view to be significant contributions to the Rashomon set workflow.

---

> > > > > ### Author Response · Authors · 2025-08-06
> > > > >
> > > > > Thank you for your response. Below is the clarification for Q1:
> > > > >
> > > > > **Q1:**  Yes, we agree. We envision an interactive scenario in which a practitioner works with a large dataset and can iteratively request the next x best models, explore them, or expand the Rashomon set until satisfied with either the number of models or the current Rashomon bound. Thank you for your comments. We will include a discussion in the paper addressing the differences between these two viewpoints on the Rashomon set.

---

> > > > > > ### Comment · Reviewer_hQ9P · 2025-08-06
> > > > > >
> > > > > > Thank you for your engagement.
> > > > > >
> > > > > > That seems like a reasonable interactive scenario, which is not too dissimilar with a similar way to use TreeFARMS -- start with a small epsilon, and increase it until satisfied with the number of models or until the bound is too large.

---

### Official Review · Reviewer_v9wz · 2025-06-24

**Clarity:** 3
**Significance:** 3
**Originality:** 2
**Rating:** 5
**Confidence:** 3

**Summary:**

This paper introduces SORTD, a method to generate Rashomon Sets of sparse decision tree models.

The proposed algorithm uses dynamic programming with bounds, caching, and specialized subroutines to efficiently represent and explore the space of sparse decision trees in order of ascending (best first) objective function. Because it generates all sparse decision trees in the Rashomon set in ascending order of their objective function, SORTD provides anytime guarantees and does not require pre-fixing the Rashomon parameter $\epsilon$ -- which are two important advances compared to the state-of-the-art.

An efficient procedure is also provided to compute partially ordered objectives in a post-hoc manner all over the Rashomon set, without having to explicitly compute it from scratch for each tree.

Thorough experiments confirm the efficiency of the proposed algorithm to generate Rashomon sets for any totally ordered objective and to evaluate them efficiently with respect to additional (partially ordered) desiderata (here, the equal opportunity statistical fairness metric).

**Questions:**

Q1/ "Our analysis indicates that this scalability stems from a compensating effect in SORTD’s search: although more features increase the number of branching nodes per search node, they also shorten the sorted solution lists each search node maintains." (line 243) -> This sentence is a bit unclear to me. Could you please clarify how increasing the number of features (so basically the number of split possibilities) can actually shorten the solution lists ?

Q2/ SORTD uses specialized subroutine for trees containing 1, 2 or 3 branching nodes, that are said to "reduce runtime and improve
scalability with both the number of features and the depth budget". Did you perform ablation experiments to precisely quantify the speedup provided by using (or not) these subroutines ?

Q3/ The fact that SORTD generates sparse decision trees within the Rashomon set in ascending order of their objective function is an important advantage as it provides nice anytime behaviour. Nevertheless, technically speaking, even though they are not generated, suboptimal solutions still need to be explored (e.g., in Figure 2, how to be sure that other splits are not better without verifying them ?). I guess this is done implicitly when calling STreeD to generate the optimal solution values of the left and right child search nodes of a given branching node ? Could you conceptually elaborate on that ?

Q4/ What would be the main challenges towards extending the proposed approach to generate Rashomon sets of other types of interpretable models ? (e.g., rule lists, whose Rashomon set was already explored and for which efficient branch-and-bound algorithms were proposed such as CORELS)

**Ethical Concerns:**

["NO or VERY MINOR ethics concerns only"]

**Final Justification:**

I maintain my original rating for this paper. As mentioned in my review, I believe this is an interesting and technically solid work. The resulting tool can be useful for the interpretable ML community. The authors' rebuttal brought some additional information regarding memory requirements and scalability that are worth including in a revised version of their paper and confirm my positive evaluation of this work.

**Limitations:**

yes

**Quality:**

3

**Strengths And Weaknesses:**

Quality: The problem is well-motivated, the related work is properly introduced and discussed. The proposed approach is technically solid, and the empirical evaluation is thorough.

Clarity: I found the paper pleasant to read. The different bounds, algorithms and subroutines are sufficiently detailed. Nice visualizations are provided for all experiments, allowing assessment of the results and their statistical significance.

 Significance: SORTD improves the state-of-the-art by (i) allowing an anytime behaviour (ii) avoiding pre-fixing a Rashomon parameter which could be too large or too small depending on the data and hypothesis class at hand and (iii) empirically improving scalability and efficiency of generating Rashomon sets for sparse decision trees (compared to state-of-the-art approaches). The ability to handle arbitrary objectives (as long as they are totally ordered and additive) is also a significant advantage of the method. I believe the provided implementation can be a useful tool for the interpretable machine learning community.

Originality: The proposed algorithm incorporates many building blocks from the literature on (optimal) sparse decision trees optimization, and on constructing their Rashomon sets. It results in a very technical contribution, with the conceptual/theoretical novelty being limited.
Nevertheless, the core idea of exploring the Rashomon set of sparse decision trees in ascending objective function order (i.e., best first) makes sense and empirically provides significant advantages (as aforementioned).

Scalability remains limited, as Rashomon sets are empirically explored only for decision trees with depth <= 5, and even in this setup, 100GB of RAM are necessary (with some experiments actually reaching the limit). But still, this is an improvement over the state-of-the-art. Maybe reporting some memory use values (in practice, are we far from using the allocated 100GB ?) could be worth it ?


Minor:
- (typo) Line 13 of Algorithm 2: what does "if sol ′.value ← next then" mean ? This looks like an assignment (with operator ←) rather than a comparison (as described in the main text on line 197). Also, "next" is undefined in this context.
- (typo) Line 923: "and a the fairness metric [...]"
- In Appendix B1, it is mentioned that 4 datasets were excluded because they "exceeded memory limits while solving". Could you clarify this ?(i.e., was the memory limit reached with the proposed SORTD, TreeFarms, or both ?)
- (typo) Line 1073: "[...], which are then evaluate [...]"
- (typo) Line 1109: "[...] objective)," (additional closing parenthesis)

---

> ### Author Rebuttal · Authors · 2025-07-31
>
> We would like to thank the reviewer for the positive comments. We reply to the questions below:
>
> **Q1:** *"...Could you please clarify how increasing the number of features ... can actually shorten the solution lists?"*
>
> We would like to highlight that the size of the Rashomon set is fixed in this analysis. Adding features would potentially increase the Rashomon set size given a fixed Rashomon bound. However, given a fixed Rashomon set size, the solutions would be distributed to the solutions list of more branching nodes.
>
> **Q2:** *"...Did you perform ablation experiments to precisely quantify the speedup provided by using (or not) these subroutines ?"*
>
> Most processes in SORTD are used to calculate the solutions in order, which are not expected to improve the runtime performance. Therefore, the scalability performance (demonstrated in Figure 3 and Figure 4) is mostly obtained through the depth-two solver.
>
> **Q3:** *"...even though they are not generated, suboptimal solutions still need to be explored (e.g., in Figure 2, how to be sure that other splits are not better without verifying them ?)..."*
>
> We manage this situation by asking for the next solutions one by one. Each branching node needs to prepare its best solution, and the search node accepts the one with the lowest value. It is possible that the best solution of a branching node is left unused; however, this would require calculating only one extra solution.
>
> **Q4:** *"What would be the main challenges towards extending the proposed approach to generate Rashomon sets of other types of interpretable models?..."*
>
> The application of the techniques of our approach to other interpretable models depends on the properties of that model. For example, can the model be solved using best-first search; can the problem be split into smaller repeated subproblems for using DP; can subproblem relations be modeled in a graph?
>
> **Weakness:** *"...Maybe reporting some memory use values (in practice, are we far from using the allocated 100GB ?) could be worth it ?"*
>
> The high memory requirements only occurred during the experiment preparation phase for precomputing the Rashomon multiplier epsilon values that were required for a fair comparison with TreeFARMS (but SORTD itself does not require them). We will clarify this in the text. The actual comparison between TreeFARMS and SORTD records no exceeded memory limits. In fact, we observe approximately 5-7 times less memory usage by our approach than TreeFARMS, and on average over all instances, SORTD uses less than 1GB of memory. We will add these results to the appendix.
>
> **Minor Weakness 2:** *"In Appendix B1, it is mentioned that 4 datasets were excluded because they "exceeded memory limits while solving. ... was the memory limit reached with the proposed SORTD, TreeFarms, or both?"*
>
> We used SORTD to precompute the precise Rashomon multiplier epsilon values for TreeFarms. This means the memory limits were observed with SORTD.
>
> **Minor Weakness 1:** *"what does "if sol ′.value ← next then" mean?"*
>
> The reviewer is correct, it is a typo. We will change the assignment operator to the equality comparison operator.

---

> > ### Comment · Reviewer_v9wz · 2025-08-01
> >
> > Thank you for your answers. Regarding the memory requirements, it is indeed a strength of your approach that would be worth clarifying in the final version of your paper.
> >
> > I maintain my score (accept) and still think this is a valuable contribution, with the resulting tool likely to be of interest to the interpretable machine learning community.

---

### Official Review · Reviewer_2J8x · 2025-06-26

**Clarity:** 4
**Significance:** 3
**Originality:** 2
**Rating:** 6
**Confidence:** 4

**Summary:**

The authors extend the STreeD decision tree optimization framework to support objective-value-ordered Rashomon set enumeration. The approach yields more than an order of magnitude improvement in runtime relative to state-of-the-art TreeFARMS, and allows extension of Rashomon sets to additive objectives with total orderings (significantly, giving us the first full Rashomon set enumeration algorithm for regression trees). The appendix details an additional benefit: the approach can compute other separable, partially ordered objectives while enumerating the Rashomon set, saving some computation time relative to computing those objectives afterwards.

**Questions:**

- Could you please expand a bit on the benefit of the depth 2 solver? My understanding from MurTree is that asymptotically this solver is quite similar to a standard brute force search unless the binarization system is particularly sparse. (i.e. if each feature has ½ of its values true and the other half false, the runtime improvement is just a factor of 4). Since this work uses a sparsity term, lambda, there’s some limit on how sparse each reasonable feature is (since there’s no reason to consider features that have fewer than 2/lambda values true for binary classification). Clearly there’s a benefit to this approach relative to TreeFARMs, but it seems like it’s not an asymptotic improvement (and indeed, it’s sort of hard for me to tell from figure 3 that the scaling with depth is better than a constant factor improved, albeit an impressive constant factor).


- How are you measuring partial progress in treefarms (i.e. how quickly each individual tree is generated), particularly when it times out? I’ve always run treefarms to completion before analyzing the set, and I’m not sure how to think about partial output from the algorithm (i.e. all the results for depth 5). It’s not clear to me from glancing at the experiment code for the treefarms comparison how intermediate results were collected, so a response here would be appreciated as it would save me some digging.

**Ethical Concerns:**

["NO or VERY MINOR ethics concerns only"]

**Final Justification:**

The authors have convincingly addressed my concerns in discussion. I'll highlight again the strengths I mentioned this work demonstrates:

-	A flexible framework for Rashomon sets across many additive objectives, including regression
-	Substantial runtime improvements
-	Giving the Rashomon set in sorted order, right away, makes finding the Rashomon set across a range of choices of epsilon more convenient.
-	I think the way of enumerating the set in objective order is quite clever, and of interest in general.

In my opinion, improving the accessibility and flexibility of a Rashomon framework for trees is a substantial contribution to the field, enough to warrant a 6. While the authors and I have a difference of opinion about how important enumerating the set in sorted order is, we certainly agree that it's a useful improvement.

**Limitations:**

I'd be interested in seeing the results in 5.2 relative to a true Rashomon set, rather than the set of top 10,000 models. This is at most a minor limitation, though.

**Paper Formatting Concerns:**

No major formatting concerns whatsoever.

Minor typo in lines 6 and 7 of algorithm 2: incrememt -> increment.

**Quality:**

4

**Strengths And Weaknesses:**

Strengths:

-	A flexible framework for Rashomon sets across many additive objectives, including regression
-	Substantial runtime improvements
-	Giving the Rashomon set in sorted order, right away, makes finding the Rashomon set across a range of choices of epsilon more convenient.
-	I think the way of enumerating the set in objective order is quite clever, and of interest in general.

Weaknesses:

-	The prior state of the art allows random sampling from the Rashomon set, which I believe this approach does not
-	While this work is correct that TreeFARMS does not preserve objective ordering, my understanding is that this work is incorrect in stating there is no ordering at all to TreeFARMS trees. TreeFARMS trees are structured in relation to the splits used, allowing users to traverse the Rashomon set by split structures as in TimberTrek [1], rather than by objective ordering (when all trees in the Rashomon set are already near-equivalent in objective value).

Minor Weaknesses:

-	While some applications clearly benefit from sorting the Rashomon set, not all do: the point of the Rashomon set for reasonable epsilon is that all models within it differ only trivially, so their relative ordering is not deeply significant. Giving a total ordering can imply meaningful differences among models that are, in practice, interchangeable.
-	With the prior state-of-the-art, it was not particularly complicated to get a top-k set of trees, or the sorted ordering, from TreeFARMS when the full set is of the size this  paper defines as reasonable (10^6 or less). The sorting can be done as post-processing in reasonable time.
-	The top 100 model approach for variable importance (section 5.2) shows some similarity to the conclusions with the top 10,000, but with limitations: a high jaccard similarity for 20 out of 29 datasets is better than the result for a single model, but still only just above 2/3 of datasets. (As a minor note - the top 100 threshold approach used here seems like it forces breaking ties arbitrarily, relative to just using a Rashomon set with a small epsilon)

[1] Wang, Zijie J., et al. "Timbertrek: Exploring and curating sparse decision trees with interactive visualization." 2022 IEEE Visualization and Visual Analytics (VIS). IEEE, 2022.

---

> ### Author Rebuttal · Authors · 2025-07-31
>
> We would like to thank the reviewer for the positive comments. We reply to the questions below:
>
> **Q1:** *"Could you please expand a bit on the benefit of the depth 2 solver?... "*
>
> The depth-two solver improves the scalability by order of magnitude in practice. The reviewer is right to observe that there is indeed no asymptotic improvement in the worst case (50% sparsity) even though the constant improvement is significant (the improvements are a function of the sparsity). Other important elements of the depth-two solver are that it re-uses the previous counts if two consecutive datasets are similar, and that it provides a strong reduction in cache entries (which contributes to a smaller memory footprint).
>
> **Q2:** *"How are you measuring partial progress in treefarms... "*
>
> We clarify that we did not evaluate the partial progress of TreeFARMS. Instead, we either waited for TreeFARMS to compute the entire Rashomon set or for it to reach the time limit and then reported the runtime. If this question refers to Figure 3, the distribution shown in the figure represents the percentage of problem instances (dataset x lambda) for which the full Rashomon set was found within the time limit, not the percentage of trees generated (we will improve the figure caption).
>
> **W1:** *"The prior state of the art allows random sampling from the Rashomon set, which I believe this approach does not"*
>
> SORTD also supports sampling. The Python interface of SORTD includes the function get_tree_n(index), which allows retrieval of a tree by its index and thus enables flexible sampling. Given SORTD’s runtime efficiency, sampling after generating the full Rashomon set can still result in a faster sampled subset compared to TreeFARMS.
>
> **W2:** *"While this work is correct that TreeFARMS does not preserve objective ordering, my understanding is that this work is incorrect in stating there is no ordering at all to TreeFARMS trees..."*
>
> In the paper we were referring to the objective ordering, we will make it clear in the paper.
>
> **MW1:** *"...Rashomon set for reasonable epsilon is that all models within it differ only trivially, so their relative ordering is not deeply significant..."*
>
>  We agree that a Rashomon set with a reasonable epsilon is valuable, but the drawback is that it requires guessing the correct epsilon. Our goal is to remove this requirement. For example, in our experiments, sometimes epsilon=10% contains only the optimal solution, and sometimes it contains millions of trees (see Table 5 and the Rashomon multipliers discussion on page 29). By asking for the top-n trees, we get more stable performance, and the anytime behavior is valuable, e.g., within a few minutes we can already get results without needing to wait for the entire Rashomon set given an epsilon. We will make this clearer in the paper.
>
>
> **MW2:** *"With the prior state-of-the-art, it was not particularly complicated to get a top-k set of trees, or the sorted ordering...The sorting can be done as post-processing in reasonable time"*
>
> That is correct, it is not complicated, but it requires guessing the correct Rashomon multiplier value and then waiting for TreeFARMS to terminate. In addition, our approach SORTD does not require epsilon, and we also compute the Rashomon set orders of magnitude faster whilst providing anytime behavior (allowing different objectives), see Figure 3 and Tables 2-4. We will include this discussion in the paper in a dedicated subsection (or Appendix).
>
>
> **MW2:** *"The top 100 model approach for variable importance (section 5.2) shows some similarity to the conclusions with the top 10,000, but with limitations..."*
>
> The purpose of our experiment in Section 5.2 and B.4 is to show that it is useful to use a small number of trees to gain insight, not necessarily to prove the true variable importance (more trees is always better in that regard). For a very large dataset such as Haberman or Biodeg, this is particularly valuable.
>
> **Limitations:** *"I'd be interested in seeing the results in 5.2 relative to a true Rashomon set..."*
>
> If we understand the ‘true’ Rashomon set as a set defined by a Rashomon multiplier epsilon, we may observe that by selecting top 10k trees, we also implicitly select an epsilon (the epsilon is defined by the last tree found). In this sense, the experiment in Section 5.2 is what reviewer is asking for. We will clarify this in the paper.

---

> > ### Comment · Reviewer_2J8x · 2025-08-03
> >
> > Thank you for taking the time to respond and further clarify the work. I have a few clarifications/comments below, in the interest of further improving an already strong paper.
> >
> > ## Clarifications/potential limitations
> > > If we understand the ‘true’ Rashomon set as a set defined by a Rashomon multiplier epsilon, we may observe that by selecting top 10k trees, we also implicitly select an epsilon (the epsilon is defined by the last tree found). In this sense, the experiment in Section 5.2 is what reviewer is asking for. We will clarify this in the paper.
> >
> > Apologies for the ambiguity with the word “true” – yes, a Rashomon multiplier is exactly what I meant. The Rashomon set in TreeFARMS (and in Breiman’s “Two Cultures” paper introducing the term) is about almost-equivalent performance rather than a set number of models (where the kth such model could be arbitrarily worse).
> >
> > **There are two significant reasons why the experiment in Section 5.2 doesn’t match this**. The first is that the Rashomon multiplier epsilon that corresponds to the top 10k trees will vary across bootstraps, instead of being a fixed value across bootstraps. In that sense, this top k method differs from a Rashomon multiplier style approach. The second is that a top-k set is not necessarily a Rashomon set for any epsilon, because there could be a tie between the element k and element k+1 (and in fact, a tie is rather likely – frequent ties in objective are noted as a key reason for TreeFARMS’ tractability in section 4 of the TreeFARMS paper, and the trend is partly explained by the fact that there are many predictively equivalent forms of the same tree in Rashomon sets[1]).
> >
> > > SORTD also supports sampling. The Python interface of SORTD includes the function get_tree_n(index), which allows retrieval of a tree by its index and thus enables flexible sampling. Given SORTD’s runtime efficiency, sampling after generating the full Rashomon set can still result in a faster sampled subset compared to TreeFARMS.
> >
> > I’ll clarify that what I mean is efficient sampling without enumerating the whole set. Is SORTD able to support a fast access to a particular index, or does it have to enumerate trees first and then access an index? If it is the latter, I’d like to direct your attention to section 4.3 of the TreeFARMS paper, which provides an excellent discussion of this:
> >
> > “If we can store the entire Rashomon set in memory, then sampling is unnecessary. However, sometimes the set is too large to fit in memory (e.g., the COMPAS data set with a regularization of 0.005 and a Rashomon threshold that is within 15% of optimal produces 10^12 trees). Our Model Set representation permits easy uniform sampling of the Rashomon set that can be used to explore the set with a much lower computational burden. Appendix C presents a sampling algorithm.”
> >
> > I suspect get_tree_n(index) may break down if the Rashomon set is of a size larger than the scope considered in this paper. I believe that is a legitimate limitation worth discussing.
> >
> > **Addressing the above issues, or explicitly discussing them in limitations, would likely increase my recommendation from accept to the highest possible score (6). Either way, I’m confident this is good work.**
> >
> > ## Suggestions on General Pitch
> > It seems that we agree (bar your concern about unpredictable Rashomon set size) on the point I raised that
> >
> > > “With the prior state-of-the-art, it was not particularly complicated to get a top-k set of trees, or the sorted ordering, from TreeFARMS when the full set is of the size this paper defines as reasonable (10^6 or less)”
> >
> > And I agree (bar the concern about ties, discussed above) with your claim that
> >
> > > “we may observe that by selecting top 10k trees, we also implicitly select an epsilon”
> >
> > Perhaps we can more generally agree that enumerating trees in sorted order up to a top k cutoff, vs finding trees up to an epsilon cutoff, is a relatively minor difference (each is implicitly doing the other, give or take some extra post-processing). That suggests to me that your paper would be strengthened by focusing on many other contributions relative to the state of the art (an excellent algorithm, incorporating the depth 2 solver, handling regression) rather than focusing on the sorting, which in my opinion is not quite as clear a strength. This is, of course, up to the authors – I state it only out of interest in clarity within the field.
> >
> > [1] McTavish, Hayden, et al. "Leveraging Predictive Equivalence in Decision Trees." arXiv preprint arXiv:2506.14143 (2025).

---

> > > ### Author Response · Authors · 2025-08-04
> > >
> > > Thank you very much for your suggestions to further improve our work. Below are our clarifications regarding the points you raised:
> > >
> > > **Variable Importance 1:** *"The first is that the Rashomon multiplier epsilon that corresponds to the top 10k trees will vary across bootstraps, instead of being a fixed value across bootstraps. In that sense, this top k method differs from a Rashomon multiplier style approach."*
> > >
> > > We agree that this is a valuable addition to the paper and will incorporate it into the text.
> > >
> > > **Variable Importance 2:** *""The second is that a top-k set is not necessarily a Rashomon set for any epsilon, because there could be a tie between the element k and element k+1"*
> > >
> > > Indeed, that could be the case. To ensure our evaluation is comprehensive, not only for the top-n trees but also for those within a specified epsilon range, we will rerun the analysis to include the same objective trees and update the results accordingly in the paper.
> > >
> > > **Sampling:** *"Is SORTD able to support a fast access to a particular index, or does it have to enumerate trees first and then access an index?"*
> > >
> > > SORTD supports fast access to a tree with a particular index. This is achieved by leveraging the solution structure of TreeFARMS (see Section A.2), which records how many unique solutions are contained within each grouped solution (i.e., Model Set). The get_tree_n() function locates the appropriate grouped solution and retrieves the corresponding tree within it. As a result, SORTD does not enumerate all individual trees unless explicitly requested for each one through functions such as get_tree_n().
> > >
> > > **Suggestions on General Pitch:** *"Perhaps we can more generally agree that enumerating trees in sorted order up to a top k cutoff, vs finding trees up to an epsilon cutoff, is a relatively minor difference (each is implicitly doing the other, give or take some extra post-processing)."*
> > >
> > > We deeply appreciate your suggestion to strengthen our overall pitch and your acknowledgment of our other contributions. However, generating the Rashomon set in order without setting (guessing) an epsilon provides anytime Rashomon set generation (specifically for large datasets), such that if we stop the search at any time, we have a complete set up to some bound. If the set is too small, we can search longer (continue where we left off), or if it is too large, we select only those bounded. This makes practical applications for large datasets much more feasible.

---

> > > > ### Comment · Reviewer_2J8x · 2025-08-04
> > > >
> > > > Thank you for the prompt response. This addresses my remaining concerns with the paper, and I'll increase my score to a 6.
> > > >
> > > > Regarding the general pitch - thank you for considering my suggestion. It's definitely acceptable to me for us to have a difference in opinion there, and you've provided a reasonable justification.
> > > >
> > > > This is excellent work that I hope to see at the conference, and am confident will have positive impacts in the field.

---

### Official Review · Reviewer_miaQ · 2025-06-30

**Clarity:** 3
**Significance:** 3
**Originality:** 2
**Rating:** 5
**Confidence:** 4

**Summary:**

The authors propose a scalable algorithm “SORTD” for enumerating the Rashomon set of sparse decision trees. (The Rashomon set is the set of models that are near-optimal, e.g., whose accuracy is within epsilon of the optimal model accuracy.) The authors build on TreeFarms, prior work that uses dynamic programming and specialized data structures to enumerate the Rashomon set of sparse trees. The distinguishing features of SORTD are that it is more scalable and it finds trees in best-to-worst order. Since trees are enumerated best-to-worst, there is “anytime” functionality (i.e., computations that only require part of the Rashomon set can be completed more quickly) and it is not necessary to know the Rashomon parameter epsilon a-priori.

**Questions:**

- The number of features correlates with runtime (Fig. 4) but looking at Table 2 in the appendix, it doesn’t totally explain the variance between datasets (nor does dataset size). Are there other dataset properties, e.g., separability or maximal accuracy, that can help explain runtime variations?
- Section 5.2 (Variable Importance Analysis) for top-5 feature stability using Jaccard and Kendall’s T, how much worse are these metrics if we use a random selection of 100 or 10000 Rashomon set members (e.g., terminating TreeFarms once the specified number of models have been sampled)? I.e., how beneficial is having the most-performant Rashomon set members for variable importance analysis on a subset of the Rashomon set?
- Any workarounds for memory constraints? (Ok if this is for future work, but what general directions are promising?)

**Ethical Concerns:**

["NO or VERY MINOR ethics concerns only"]

**Final Justification:**

It's a well written paper that provides a solid technical contribution with potential for high impact for practitioners and researchers. The rebuttal addressed all my concerns.

**Limitations:**

Mostly — see comments about scaling and memory concerns

**Quality:**

3

**Strengths And Weaknesses:**

###Strengths
* Potential for high impact, because the code is available as a python package and this paper improves on the scalability bottleneck of existing methods for Rashomon set enumeration.
* The experiments are comprehensive
* As described in section 5.3 (regression trees and fairness metrics), the  approach can be used for real-world important tasks


###Weaknesses
* Inherent limitations to the approach include that it is limited to sparse decisions trees and it has high memory requirements. I think the high memory requirements is of particular concern because it would limit how widely this approach can be used.
* Minor: is there a typo in Fig 2? (Swap lindex and rindex in “next” and “candidates queue”)

---

> ### Author Rebuttal · Authors · 2025-07-31
>
> We would like to thank the reviewer for the positive comments. We reply to the questions below:
>
> **W1 & Q3:**  *"...I think the high memory requirements is of particular concern because it would limit how widely this approach can be used..."*,  *"Any workarounds for memory constraints?..."*
>
> The high memory requirements only occurred during the experiment preparation phase for precomputing the Rashomon multiplier epsilon values that were required for a fair comparison with TreeFARMS (but SORTD itself does not require them). We will clarify this in the text.The actual comparison between TreeFARMS and SORTD records no exceeded memory limits. In fact, we observe approximately 5-7 times less memory usage by our approach than TreeFARMS, and on average over all instances, SORTD uses less than 1GB of memory. We will add these results to the appendix.
>
> The memory requirements could potentially be further decreased by
>
> + Wiping the cache (partly) at least for the suboptimal solutions that are never used in the Rashomon set [1], which could additionally contribute to the runtime performance,
> + Gradually increasing the Rashomon multiplier until a certain set size is obtained,
> + Keeping track of fewer statistics, or only those we are interested in because our experiments were run while SORTeD was using various data structures to calculate statistics such as “number of trees with feature f”.
>
> **Q1:** *"...Are there other dataset properties, e.g., separability or maximal accuracy, that can help explain runtime variations?"*
>
> Other dataset properties that help explain runtime variations are (i) separability, (ii) maximal accuracy, (iii) equivalent feature vectors, (iv) feature vector sparsity, (v) feature similarity, among others. For example, if a 100% accurate tree exists for a given depth limit, the phase 1 (identifying the optimal tree and setting the Rashomon bound) search can terminate early. If not, much more time is spent proving that the current best tree is indeed optimal.
>
> **Q2:** *"(Variable Importance Analysis)...how much worse are these metrics if we use a random selection of 100 or 10000 Rashomon set members (e.g., terminating TreeFarms once the specified number of models have been sampled)?..."*
>
> A randomly selected set of trees might provide more stable variable importance estimates than a single tree; however, the runtime efficiency of SORTD remains essential for solving such difficult instances. In our experiments, we observed that TreeFARMS failed to find optimal solutions for some problem instances when the dataset had high dimensionality and the cost-complexity parameter $\lambda$ was low (e.g., the Haberman dataset with $\lambda=0.001$).
>
>
> **W2:** *"Minor: is there a typo in Fig 2? (Swap lindex and rindex in “next” and “candidates queue”)"*
>
> Yes, we will correct it, thank you.
>
>
> [1] Aglin, G., Nijssen, S., & Schaus, P. (2023). Learning optimal decision trees under memory constraints. In I. Ntoutsi, C. Tintarev, G. Vretos, A. S. Verebes, & D. Frossard (Eds.), Machine Learning and Knowledge Discovery in Databases. ECML PKDD 2023. Lecture Notes in Computer Science (Vol. 14139, pp. 393–409). Cham: Springer.

---

> > ### Comment · Reviewer_miaQ · 2025-08-01
> >
> > thanks, the clarifications are helpful! keeping my score at 5 (accept)

---

### Official Review · Reviewer_TncC · 2025-07-02

**Clarity:** 3
**Significance:** 3
**Originality:** 2
**Rating:** 4
**Confidence:** 4

**Summary:**

Rashomon sets of sparse decision trees are useful for downstream data analysis, such as variable importance analysis. However, getting the entire Rashomon set might be too time-consuming. To circumvent this difficulty, the authors propose to only get the top-$n$ sparse trees, using a best-first search approach so that the algorithm can exit early. To further improve the computational efficiency, the authors apply the depth-two subroutine, which has already been explored in previous works. Experimental results show that the proposed method, SORTeD, can run significantly faster than the SOTA TreeFARMS method.

**Questions:**

1. Can you confirm whether my understanding (alternating between popping and exploring) is correct?
2. Can you perform an ablation study by only using best-first search but NOT using depth-two routine? How does this compare with SORTeD in terms of running time?
3. The time limit (300) is too small in my opinion. The idea of collecting the Rashomon set is to spend some time collecting the entire set. We only do this once and save all the trees. Afterwards, we can explore the entire space instantaneously by making queries in this Rashomon set space. In this sense, it is fine if practitioners can spend some time waiting for collection process to finish. Could you increase the running time (300-> 3600) and do the time comparison again? I am interested in seeing whether TreeFAMRS can at least do a descent job (by finish collecting all the trees in the Rashomon set or the top-$n$ trees) if practitioners are willing to wait.
4. For variable importance analysis, it's unclear to me whether using the top-$n$ trees and all trees in the Rashomon set provide the same results. Can you elaborate on this? Can top-$n$ trees provide the same quantitative and qualitative variable importance analysis? Under what circumstances? Can we get any theories in this direction? If these two approaches give different variable importance analysis results, then it's unclear what benefit we get by obtaining only the top-$n$ trees. This is actually an important question. If getting the top-$n$ trees is not enough, then it undermines the motivation of this work.
5. Unlike Figure 3, Figure 4 only contain solid lines. What happens when $n=10^4$ and $n=10^6$ for Figure 4?

**Ethical Concerns:**

["NO or VERY MINOR ethics concerns only"]

**Final Justification:**

In summary, I think the key contribution of this paper is proposing to incorporate the depth-two solver into the branch and bound framework to make the overall BnB run fast. We can regard this as a new heuristic technique in finding good solutions when the tree depth is small. TreeFARMS didn't spend much time in designing good heuristics and its past series of papers focus more on deriving the lower bounds for pruning purposes. It's intuitively understandable now why the author claims their work is a step further than TreeFARMS.

**Limitations:**

There isn't a specific paragraph discussing the limitations in the main paper. However, such a discussion is included in one of the checklist answers.

**Quality:**

3

**Strengths And Weaknesses:**

**Strength**

1. The writing for the introduction paragraph is pretty strong. The language is both concise and comprehensive.
2. Clear run-time improvement over the baseline TreeFARMS.
3. The main contribution is an engineering effort in C++ programming to make the whole program work for best-first search on the task of collecting the Rashomon set. Conceptually, the contribution is fair (See my #5 below for the weaknesses)

**Weakness**

3. The writing for the methodology section (Section 4) is a little bit too complicated in my opinion. I spent several hours without understanding in details of what is going on. The central idea is simple. The first phase is popping. We keep popping out the best solution in the queue. If we cannot certify the solutions in the queue are the best, we should switch to the second phase --- exploring (to certify optimality in the remaining search space). We alternate between the popping phase and exploration phase until we reach the Rashomon set limit. I suggest the authors give this big picture at the beginning of Section 4.

4. To continue on this thread, I suggest the authors highlight/boldface this sentence "If the child search node does not have that solution yet, it is obtained by calling GetNextSolution (Alg. 1)." I believe this is the key step for understanding the big picture. We need to alternate between popping and further exploration.

5. Conceptually, there aren't any big surprising innovations. Best-first search is one of the standard searching methods. Depth-two subroutine has already been proposed in previous works.

6. This is a minor comment. In some sense, the substantial time-improvement isn't that too surprising. Firstly, the goals of SORTeD and TreeFARMS are different. SORTeD only care about the top-$n$ trees in the Rashomon set, while TreeFARMS want to collect ALL the trees in the Rashomon set. Secondly, SORTeD use the depth-two subroutine, which I believe play a big part in the time improvement. It is worthwhile to conduct some ablation study to see how much an impact this depth-two subroutine play by turning it off and just having the best-first search framework.

---

> ### Author Rebuttal · Authors · 2025-07-31
>
> We would like to thank the reviewer for the positive comments. We reply to the questions below:
>
> **Q1:** *"Can you confirm whether my understanding (alternating between popping and exploring) is correct?"*
>
> It is correct that popping and exploring procedures are alternatingly called.  However, the solutions that are inserted into the sorted solution lists are guaranteed to be the best. Popping continues until all solutions in the sorted solution list are used and the next one is requested.  In that case, exploration (GetNextSolution function call) begins.
>
> **W3 \& W4:**  *"The writing for the methodology section (Section 4) is a little bit too complicated...I suggest the authors give this big picture at the beginning of Section 4."*
>
> Thank you for your suggestions. We will provide the big picture both for the search node and for the helper nodes so that it is easier to understand their roles and how these roles are linked to each other.
>
> We will additionally highlight the sentence *"If the child search node does not have that solution yet, it is obtained by calling GetNextSolution (Alg. 1)"*.
>
> **W5:** *"Conceptually, there aren't any big surprising innovations. Best-first search is one of the standard searching methods..."*
>
> Due to the following reasons, we consider our methodology to be innovative:
>
> + Although best-first search is a well-known method, to the best of our knowledge, our application of BFS in an and/or-search graph that yields lists of solutions by computing a Cartesian sum of (yet incomplete) subproblem solution lists is novel (see lines 155-157).
> + We contribute a new gradual solution creation technique for the depth-two solver (Appendix A.7) to improve its anytime performance.
> + Unlike previous work, SORTD can compute and evaluate Rashomon sets across a range of tasks, including classification and regression objectives.
>
> **Q2 & W6b:** *"Can you perform an ablation study by only using best-first search but NOT using depth-two routine?...", "...SORTeD use the depth-two subroutine, which I believe play a big part in the time improvement..."*
>
> The scalability advantage is mostly obtained through the depth-two solver. Our specific contribution in this component is the use of a gradual solution creation technique (Appendix A.7), which improves its anytime performance. The novel search technique does not contribute to SORTD’s runtime but provides an anytime behavior, which we believe is essential to parse larger datasets, specifically, if good Rashomon bounds cannot be guessed beforehand.
>
> **W6a:** *"...In some sense, the substantial time-improvement isn't that too surprising. Firstly, the goals of SORTeD and TreeFARMS are different..."*
>
> + Our approach solves the same problem as TreeFARMS, i.e., given some bound, return the whole Rashomon set. Our approach supports setting this bound in the same way TreeFARMS does: by some relative performance to the optimal solution (implied by the Rashomon multiplier epsilon). In addition, we also support setting the bound by fixing the size of the Rashomon set.
> + We ensure a fair comparison by running TreeFARMS with precise epsilon input such that it outputs the same number of trees as SORTD. Under this fair comparison, SORTD has a substantial runtime improvement over TreeFARMS.
>
>
> **Q3:** *"The time limit (300) is too small..."*
>
> Our approach is two orders of magnitude faster and offers anytime behavior, as shown in Fig. 3 and Tables 2–4.  In practice, this allows SORTD to be used for exploring larger datasets within a similar time limit. While we could extend the time limit, the benefit of doing so is unclear to us.
>
> **Q4** *"For variable importance analysis, it's unclear to me whether using the top- trees and all trees in the Rashomon set provide the same results... If getting the top- trees is not enough, then it undermines the motivation of this work."*
>
> We want to correct the misunderstanding that the difference is between evaluating the top-n trees and all trees in the Rashomon set. Instead, the difference is how the Rashomon set is defined: by its size (top-n) or by some Rashomon multiplier epsilon.The purpose of our experiment in Section 5.2 and B.4 is to show the variable importance measured is relatively stable for 100 or 10,000 trees. Developing theories on the necessary number of trees, rather than the bound, to get stable measurements of variable importance is an interesting future work.
>
> Beyond facilitating variable importance analysis, the sorted Rashomon set also aids the model selection process by generating trees in order while requiring less runtime.
>
> **Q5:** *"Unlike Figure 3, Figure 4 only contain solid lines. What happens when  and  for Figure 4?"*
>
> In Figure 4, we show the cumulative runtime distribution across datasets with $n^T = 10^6$ and maximum depth 4. We will make this clear in the figure caption and explanation.  The figure shows that as the number of features increases, the proportion of problem instances for which the Rashomon set calculation finishes within the time limit decreases. This effect is notably more pronounced for TreeFARMS compared to SORTD.
>
> **Limitations:** *"There isn't a specific paragraph discussing the limitations in the main paper. However, such a discussion is included in one of the checklist answers."*
>
> That is a good point, we will provide the limitations in the main paper.

---

> > ### Comment · Reviewer_TncC · 2025-08-06
> >
> > Thanks for your answers. I want to ask some clarifying questions and make some comments.
> >
> > 1. ```The scalability advantage is mostly obtained through the depth-two solver.``` Since I asked for an ablation study during my initial review, and there wasn't one provided, can I say that without using this depth-two solver, the codebase will run the same time or potentially slower than TreeFARMS? Otherwise, I'd assume the ablation study results would be reported here for rebuttal.
> > 2. What does this sentence mean? ```The novel search technique does not contribute to SORTD’s runtime but provides an anytime behavior```. Does this mean that if I set epsilon=0.5% and the corresponding total Rashomon set is 10^3, and we set number of top trees I want to require is n=10^4, does your codebase run on the same time scale as TreeFARMS? In other words, your codebase is only significantly faster if n << total number of trees in the entire Rashomon set?
> > 3. ```While we could extend the time limit, the benefit of doing so is unclear to us.``` That's exactly why I'm asking the question regarding variable importance. Once we have the entire Rashomon set, we can conduct variable importance. It's much unclear to me whether only obtaining the top-n trees (when n is small) provide the same variable importance analysis results as using a large Rashomon set. For example, in [Rui Xin et. al's paper](https://proceedings.neurips.cc/paper_files/paper/2022/file/5afaa8b4dd18eb1eed055d2d821b58ae-Paper-Conference.pdf), on bottom of page 7, they stated that they can find 10^8 trees on the Monk2 dataset. Significantly large Rashomon set provide important information, it tells us which features are repeatedly used and which features are much less frequently used. The frequency of features appearing in different trees of the Rashomon set is by itself a valuable pattern of the dataset. You can do uncertainty quantification, feature distribution analysis, etc. If the Rashomon set is small, we are only capturing a point estimate in a statistical sense, NOT the distribution of the variable.
> > 4. Question: in Fig. 3 and Tables 2–4, are you getting the top-n trees of the Rashomon set, where n is less than the total number of trees in the Rashomon set (defined by the tolerance epsilon) or keep increasing n until it is bigger than or equal to the total number of trees in the Rashomon set? Based on your rebuttal answer, I assume the total number of the trees in the Rashomon set will never be bigger than 10^6? This is two orders of magnitude less 10^8 on the Monk2 dataset used in Rui Xin's paper, as I have mentioned.
> > 5. Question: continuing from the previous question. if your n is never bigger than the total number of trees in the Rashomon set (meaning the total number of trees in the Rashomon set is greater than 10^6), how do you calculate the total computed (%) on the y-axis in Figure 3? If I understand correctly, you are only reporting the percentage of all experiments for which the algorithm terminates under the running time limit right? You are not reporting how many trees each algorithm collect during that time. TreeFARMs could run much longer, due to the fact that its heuristic search is not based on best-first search, but when TreeFARMS terminates, it gets far more number of trees from the Rashomon set.
> > 6. For your Table 2-4 results, the running time are reported in an aggregated way (with mean and variance) for all n={10^1, 10^2, ..., 10^6}. This is not an intuitive way to report running time for different scales of n. It could be that when n is small, your best first search based method terminates early and TreeFAMRS is still collecting more trees while when n is large, the two methods run comparably. Therefore, all the advantages shown here is due to the fact of early termination, which lose its benefit when we go to large n, such as 10^8. A better way to report is report run time for different n separately.
> > 7. As I've said, the point of collecting the Rashomon set is not as much about ranking optimal trees on the fly (in the online fashion as you have done here; I think your terminology is "anytime performance") as about getting a huge Rashmon set, which will by itself be valuable for lots of downstream tasks. It is totally okay to let the algorithm run minutes, hours, or overnight to collect a huge set of trees. After we collect all these trees, ranking and sorting them by loss or some other metrics is trivial and can be done instantaneously.
> > 8. Continuing from the previous point, if I look at your Figure 3, the total running time limit is 300 seconds. While you can say that your algorithm is 1-2 orders of magnitude faster than TreeFARMS, the time scale we are looking at is 30s -> 3s or 0.3s. The contribution will be much more valuable if for a huge Rashomon set, we can reduce the running time from 5 hours to 2-3 minutes. That's why I'm insisting to request additional results on setting the running time limit to 3600s or setting n=10^8 or larger.

---

> ### Author Response · Authors · 2025-08-08
>
> Thank you for your suggestions to improve our work. Below are our clarifications regarding the points you raised:
>
> **1:** *"...can I say that without using this depth-two solver, the codebase will run 	the same time or potentially slower than TreeFARMS?"*
>
> Our depth-two solver is a key element for scalability, so without it, SORTD would have the same or potentially slower runtime than TreeFARMS.
>
> **2.1:** *‘What does this sentence mean? “The novel search technique does not contribute to SORTD’s runtime but provides an anytime behavior.” ’*
>
> We were commenting on the fact that we provide trees in sorted order of their objective such that stopping the search at any point yields a Rashomon set bounded by the objective of the last returned tree. This is not a trick to improve runtime, but a useful feature. Runtime improvement is obtained through the depth-two solver.
>
> **2.2:** *"... your codebase is only significantly faster if n << total number of trees in the entire Rashomon set? "*
>
> No, SORTD is significantly faster than TreeFARMS while computing the same number of trees, i.e, the whole Rashomon set defined by the Rashomon multiplier epsilon (Sections 5.1 and B.2). To ensure a fair comparison with TreeFARMS, we precompute the required epsilons to get $10^n, n \in \{1, ..., 6 \}$ with SORTD, see Table 5 in the supplementary material for some examples.
>
> **3.1:** *"It's much unclear to me whether only obtaining the top-n trees (when n is small) provide the same variable importance analysis."*
>
> While we see your point, our intention in Section 5.2 is to highlight the potential of exploring variable importance across different Rashomon set sizes. We will make it clearer in the paper.
>
> **3.2:** *‘ “While we could extend the time limit, the benefit of doing so is unclear to us.” That's exactly why I'm asking the question regarding variable importance.’"*
>
> We clarify that even if TreeFARMS were given more time and completed the Rashomon set computation, it would still return the same number of trees as SORTD, since both methods were run with the same epsilon values in our experiments.
>
> **4.1:** *“...are you getting the top-n trees of the Rashomon set, where n is less than the total number of trees in the Rashomon set ... or keep increasing n until it is bigger than or equal to the total number of trees...”*
>
> See our answer to Q2 above; in short, in the comparison, both SORTD and TreeFARMS solve exactly the same problem, i.e., both using the same epsilon (we do not use our top-k functionality in this comparison).
>
> **4.2:** *“... 10^8 on the Monk2 dataset used in Rui Xin's paper..”*
>
> Yes, it is possible to obtain $10^8$ or more trees using the Monk2 dataset. For example, with $\lambda=0.001$, $\varepsilon=0.1$ and maximum depth 6 both methods obtain $2 \cdot 10^{12}$ trees. However, TreeFARMS takes 28 seconds whereas SORTD takes less than a second.
>
> **5.1 & 6.1:** *“TreeFARMs could run much longer ... when TreeFARMS terminates, it gets far more number of trees...”, "It could be that when n is small, your best first search based method terminates early and TreeFAMRS is still collecting more trees..."*
>
> See our answer to Q2 above; in short, since we solve exactly the same problem, both methods collect the same number of trees, and the observed advantage is purely because of our scalability improvements rather than early termination.
>
> **5.2 & 6.2:** *“... you are only reporting the percentage of all experiments for which the algorithm terminates under the running time limit right?”, “A better way to report is report run time for different n separately.”*
>
> Fig. 3 indeed reports the percentage of problem instances (dataset x configuration) solved within the given runtime. Each line represents the distribution of problem instances for which the complete Rashomon set has $10^n$ trees. Therefore, we report results without aggregation for $10^n$ in Fig. 3.
>
> **7:** *“It is totally okay to let the algorithm run minutes, hours, or overnight to collect a huge set of trees.”*
>
> While we do not necessarily disagree with you, we would like to highlight that compared to TreeFARMS, SORTD computes the Rashomon set orders of magnitude faster, scales better with large datasets, supports a variety of objectives, and does so in an anytime fashion. We see this as an advantage.
>
> **8:** *“I'm insisting to request additional results on setting the running time limit to 3600s or setting n=10^8 or larger.”*
>
> We see your point; the goal is to further emphasize the runtime advantage of our method. We promise to do the following. We will take datasets and configuration from Tables 2-4 (Appendix) where TreeFARMS took more than 300 seconds to compute, whereas our approach took seconds. We will then run TreeFARMS for an hour and report the runtime difference in a separate section in the Appendix. We agree that this provides further support for our SORTD method.

---

### Decision · Program_Chairs · 2025-09-17

**Decision:**

Accept (spotlight)

**Comment:**

This study introduces a novel anytime algorithm for enumerating Rashomon sets of sparse decision trees across various objective functions. The paper has received five reviews, with scores ranging from weak to strong acceptance. Reviewers unanimously agree on the paper's quality, its positioning in relation to related work, the benefits of this anytime enumeration method, and the significance of the experimental results. In summary, this represents an important contribution to the field of interpretable machine learning and the enumeration of Rashomon sets. For these reasons, I strongly recommend its acceptance.

Additionally, the authors effectively addressed several clarity issues during the rebuttal phase. For the revised version of the paper, I suggest incorporating this discussion to further enhance its quality.